# Addressing Misspecification in Simulation-based Inference through Data-driven Calibration

## Abstract

Driven by steady progress in generative modeling, simulation-based inference (SBI) has enabled inference over stochastic simulators. However, recent work has demonstrated that model misspecification can harm SBI's reliability, preventing its adoption in important applications where only misspecified simulators are available. This work introduces robust posterior estimation (RoPE), a framework that overcomes model misspecification with a small real-world calibration set of ground truth parameter measurements. We formalize the misspecification gap as the solution of an optimal transport problem between learned representations of real-world and simulated observations, allowing the method to learn a model of the misspecification without placing additional assumptions on its nature. The method shows how a small calibration set can be leveraged to offer a controllable balance between calibrated uncertainty and informative inference even under severely misspecified simulators. Our empirical results on four synthetic tasks and two real-world problems with ground-truth labels demonstrate that RoPE outperforms baselines and consistently returns informative and calibrated credible intervals.

## 1 Introduction

Many fields of science and engineering have shifted in recent years from modeling real-world phenomena through a few equations to relying instead on highly complex computer simulations. While this shift has increased model versatility and the ability to explain or replicate complex phenomena, it has also necessitated the development of new statistical inference methods. In particular, state-of-the-art simulation-based inference (SBI, Cranmer et al., 2020) algorithms leverage neural networks to learn surrogate models of the likelihood (Papamakarios et al., 2019), likelihood ratio (Hermans et al., 2020), or posterior distribution (Papamakarios & Murray, 2016), from which one can extract confidence or credible intervals over the parameters of interest given an observation. While SBI has proven helpful when the simulator is a faithful description of the studied phenomenon, e.g., for scientific applications (Delaunoy et al., 2020; Brehmer, 2021; Lückmann, 2022; Linhart et al., 2022; Hashemi et al., 2022; Tolley et al., 2023; Avecilla et al., 2022), recent work has also highlighted the unreliability of SBI methods under model misspecification (Cannon et al., 2022; Schmitt et al., 2023) common in many settings, thereby limiting their applicability. As a remark, our usage of the term *calibration* refers to *labeled real-world* observations and should not be confused with its usage in the context of model mis-calibration in well-specified SBI (Hermans et al., 2022), as further discussed in Appendix A

**Addressing Misspecification with a Calibration Set.** We are motivated by the potential of SBI in important applications where (1) the goal is to estimate a hard-to-measure variable from indirect but readily available measurements of other variables, but (2) only misspecified simulators relating them are available. For example, inferring properties of a patient's cardiovascular system—that can only be invasively measured—from non-invasive and abundant measurements of other physiological signals (Wehenkel et al., 2023). Or, the development of soft sensors to monitor industrial processes in real-time, where directly measuring the quantity of interest is costly and time-consuming—e.g., through laboratory analysis—but where related variables can be quickly and inexpensively measured (Jiang et al., 2021; Perera et al., 2023). For such settings, practitioners—e.g., doctors performing a diagnosis or operators of a chemical plant—will not trust the output of a method without first verifying its accuracy on a validation set with ground-truth labels. A few observations from this set

can be used as a calibration set for methods such as ours. Hence, in this work, we focus on extending SBI methodology to such applications and address model misspecification through a calibration set consisting of only a few pairs of real-world observations and their corresponding ground-truth labels. Therefore, our method does not apply to settings where SBI is used to infer non-measurable parameters, as this precludes the existence of a calibration set.

**Misspecification in SBI.** A model is a simplified description of a real-world phenomenon that allows reasoning about its properties. In the context of SBI, the model is a simulator $p(\mathbf{x_s} \mid \theta)$ that relates a parameter of interest $\theta \in \Theta$ to a distribution of simulated observations $\mathbf{x}_s \in \mathcal{X}$. In the Bayesian inference literature (Walker, 2013), the model is said to be misspecified with respect to some true data-generating process $p^\star$ producing i.i.d. real observations $\mathbf{x}_o \sim p^\star$, if the latter does not fall within the family of distributions defined by the model, i.e. $\nexists \theta \in \Theta : p(\cdot \mid \theta) = p^\star$. Based on this definition, model misspecification in both likelihood-based and simulation-based inference settings has gained a lot of interest from the research community. Among developed strategies, works that take inspiration from generalized bayesian inference (Bissiri et al., 2016) are numerous (Dellaporta et al., 2022; Chérief-Abdellatif & Alquier, 2020; Matsubara et al., 2022; Pacchiardi & Dutta, 2021; Schmon et al., 2020; Gao et al., 2023; Frazier et al., 2023). In the specific context of SBI, recent works (Ward et al., 2022; Huang et al., 2023; Kelly et al., 2023) have investigated solutions to improve the robustness of existing neural-network-based SBI methods to model misspecification and to detect it at inference time (Schmitt et al., 2023). Similarly, Frazier et al. (2020) studied the impact of model misspecification on approximate Bayesian computation methods (ABC, Rubin, 1984), introducing diagnostics to detect it and proposing strategies to make ABC robust. For the interested reader, Nott et al. (2023) review restricted likelihood methods, Bayesian modular inference, and parametric projection methods, which are standard frameworks to handle model misspecification in likelihood-based Bayesian inference.

While a source of inspiration to this work, these works do not provide direct solutions to the problem setting we are interested in, described in the second paragraph of the introduction. For the settings we consider, our simulator models the relationship between the real observations $\mathbf{x}_o$ and the parameters of interest $\theta$ *as they appear in the calibration set*. Therefore, the standard definition is insufficient, as a model may be well-specified but still yield incorrect credible intervals for the parameters of interest $\theta$; we provide an illustrative example in Appendix A. To address this issue, we define model misspecification differently. First, we assume the calibration set $\{(\theta^i, \mathbf{x}_o^i)\}_{i=1}^{N_c}$ of real-world observations $\mathbf{x}_o \in \mathcal{X}$ and their corresponding labels $\theta \in \Theta$ are sampled i.i.d. from a joint distribution given by the density $p^\star(\theta, \mathbf{x}_o)$. Let $p^\star(\theta)$ be the marginal density of the underlying parameters $\theta$ in the real world, and $p^\star(\mathbf{x}_o) := \int_\Theta p^\star(\theta) p^\star(\mathbf{x}_o \mid \theta) d\theta$ be the marginal density of the real-world observations, where $p^\star(\mathbf{x}_o \mid \theta)$ is the unknown process which is modeled by the simulator, whose implicit likelihood is denoted $p(\mathbf{x}_s \mid \theta)$. We say the simulator is misspecified if $\exists \mathcal{S} \subseteq \Theta \times \mathcal{X}$ with $\iint_\mathcal{S} p^\star(\theta, \mathbf{x}) d\theta d\mathbf{x} > 0$ such that $p^\star(\mathbf{x}_o \mid \theta) \neq p(\mathbf{x}_s = \mathbf{x}_o \mid \theta)$ for all $(\theta, \mathbf{x}_o) \in \mathcal{S}$. In this context, even if the prior distribution $p(\theta)$ is well-specified, i.e., $p(\theta) = p^\star(\theta)$, the posterior distribution obtained from the simulator would yield to inaccurate parameter predictions.

**Our Contributions.** We introduce robust posterior estimation (RoPE), an algorithm that addresses model misspecification to provide accurate uncertainty quantification for the parameters of black-box simulators. The main challenge of a misspecified setting lies in the absence of a paired datasets of simulated and corresponding real outputs. To handle this knowledge gap, RoPE proposes to estimate (using samples) a coupling between real $\mathbf{x}_o$ and simulated $\mathbf{x}_s$ observations using optimal transport (OT, Peyré et al., 2017; Villani et al., 2009). In addition to such a coupling, we consider a realistic scenario where, to improve, performance, RoPE also has access to a small, real-world calibration set of paired parameters and observations. The algorithm extends neural posterior estimation (Papamakarios & Murray, 2016) and models misspecification using OT. We evaluate the performance of the algorithm on existing benchmarks from the SBI literature, and introduce four new benchmarks, of which two are synthetic and two come from real physical systems for which both labeled data and simulators are available. To the best of our knowledge, the latter constitute the first real-world benchmarks that directly provide a ground truth for the inferred parameters for SBI under misspecification. We perform additional experiments to explore the effect that different calibration set sizes, prior misspecification, and distribution shifts have on the performance of the algorithm, together with ablation studies to understand the impact of each of its components.

## 2 BACKGROUND & NOTATION

In this section, we provide a short review of SBI and OT, as our method is at the intersection of these two fields. We start with some fundamental definitions. We consider a simulator, implemented as a computer program $S : \mathbb{R}^K \times [0,1] \to \mathbb{R}^D$, that takes in physical parameters $\theta \in \Theta \subseteq \mathbb{R}^K$ and a random seed $\varepsilon \in [0,1]$ to generate measurements $\mathbf{x}_s \in \mathcal{X}_s \subseteq \mathbb{R}^D$. The simulator is a simplified version of a real and unknown generative process $\mathbb{P}^\star$ that produces real-world observations $\mathbf{x}_o \in \mathcal{X}_o \subseteq \mathbb{R}^D$. We assume this process depends on parameters with the same physical meaning as the ones of the simulator and thus use the same notation $\theta$. Our goal is to estimate a well-calibrated and informative posterior distribution $p(\theta \mid \mathbf{x}_o^i)$ for each observation in the test set $\mathbf{x}_o^i \in \mathcal{D}$, which reduces uncertainty compared to the prior distribution. As a remark, the most informative and calibrated posterior is the Bayesian posterior $p^\star(\theta \mid \mathbf{x}_o)$ that corresponds to the true generative process $p^\star(\mathbf{x}_o) := \int p^\star(\mathbf{x}_o \mid \theta) p^\star(\theta) \mathrm{d}\theta$. To achieve our goal, we have access to **1.** the misspecified simulator $S$ that embeds domain knowledge and approximates $p^\star(\mathbf{x}_o \mid \theta)$, **2.** a small calibration set of labeled real-world observations $\mathcal{C} := \{(\theta^i, \mathbf{x}_o^i)\}_{i=1}^{N_c}$, which enables data-driven correction of the simulator's misspecification, **3.** a test set $\mathcal{D} := \{\mathbf{x}_o^i\}_{i=1}^{N_o}$ of real-world observations arising from $\mathbb{P}^\star$ for which we want to estimate the posterior, and **4.** a prior $p(\theta)$ that approximates the marginal distribution $p^\star(\theta)$ of parameters in the real-world.

### 2.1 SIMULATION-BASED INFERENCE (SBI)

Applying statistical inference to simulators is challenged by the absence of a tractable likelihood function (Cranmer et al., 2020). As a solution, SBI algorithms leverage modern machine learning methods to tackle inference in this likelihood-free setting (Lueckmann et al., 2021; Delaunoy et al., 2021; Glöckler et al., 2022). Among SBI algorithms, neural posterior estimation NPE (Papamakarios & Murray, 2016; Lueckmann et al., 2017; Radev et al., 2020) is a broadly applicable method that trains a conditional density estimator of $p(\theta \mid \mathbf{x}_s)$ from a dataset of parameter-simulation pairs. In this paper, we focus on making NPE robust to model misspecification.

NPE usually parametrizes the posterior with a neural conditional density estimator (NCDE), which is composed of (1) a neural statistic estimator (NSE), denoted by $\mathbf{h}_\omega : \mathcal{X}_s \to \mathbb{R}^l$, that compresses observations into $l$-dimensional representations and, (2) a normalizing flow (NF, Papamakarios et al., 2021; Tabak & Vanden-Eijnden, 2010) that parameterizes the posterior density as $p_\phi(\theta \mid \mathbf{h}_\omega(\mathbf{x}_s))$. The parameters $\phi$ and $\omega$ of the NCDE are trained with stochastic gradient ascent on the expected log-posterior probability, solving the following optimization problem

$$\phi^\star, \omega^\star = \arg\max_{\phi, \omega} \mathbb{E}_{\substack{\theta \sim p(\theta) \\ \varepsilon \sim \mathcal{U}[0,1]}} \left[ \log p_\phi(\theta \mid \mathbf{h}_\omega(S(\theta, \varepsilon))) \right], \tag{1}$$

where $p(\theta)$ denotes a prior distribution over the parameters $\theta$.

Under the assumption that the class of functions represented by the NCDE contains the true posterior, solving (1) leads to a perfect surrogate $p_{\phi^\star}(\theta \mid \mathbf{h}_{\omega^\star}(\mathbf{x}_s))$ of the true posterior $p(\theta \mid \mathbf{x}_s)$. In that case, $\theta \perp \mathbf{x}_s \mid \mathbf{h}_{\omega^\star}(\mathbf{x}_s)$, that is, the NSE $\mathbf{h}_{\omega^\star}$ is a sufficient statistic of $\mathbf{x}_s$ for the parameter $\theta$ (Chen et al., 2020; Wrede et al., 2022; Chan et al., 2018). In practice, we approach perfect training by generating a sufficiently large number of pairs $(\theta, \mathbf{x}_s)$ and doing a search on the NCDE's architecture and training hyperparameters. To simplify notation, we denote the NCDE learned with NPE as $\tilde{p}(\theta \mid \mathbf{x}_s)$

### 2.2 SEMI-BALANCED OPTIMAL TRANSPORT (OT)

As detailed in Section 3, RoPE models the misspecification between simulations and real-world observations as an OT coupling. For readers unfamiliar with OT, an OT coupling is a mathematical object that represents the most efficient way to associate two probability distributions, i.e., minimizing a cost function that measures the "distance" between samples drawn from each distribution. The cost function $c : \mathcal{X}_o \times \mathcal{X}_s \to \mathbb{R}$ assigns a cost to any pair $(\mathbf{x}_o, \mathbf{x}_s) \in \mathcal{X}_o \times \mathcal{X}_s$.

In our setting, we can access a limited number $N_o$ of real-world observations $\{\mathbf{x}_o^i\}_{i=1}^{N_o}$, which we assume result from an unknown generative process $p^\star(\mathbf{x}_o) = \int p^\star(\mathbf{x}_o \mid \theta) p^\star(\theta) \mathrm{d}\theta$. Writing $C = [c(\mathbf{x}_o^i, \mathbf{x}_s^j)]_{ij}$ for the cost matrix between observed and simulated data, we solve the discrete semi-balanced (Rabin et al., 2014) entropy-regularized (Frogner et al., 2015) OT problem to recover a flexible coupling that is constrained to match the observed points but has the flexibility to discard simulated points. Namely, given a set $\{\mathbf{x}_s^j\}_{j=1}^{N_s}$ of simulated observations, we search for the non-

negative transport matrix $P^\star$ that satisfies its left marginal constraint,

$$\mathcal{B}_o = \left\{ P \in \mathbb{R}_+^{N_o \times N_s} : \sum_{j=1}^{N_s} P_{ij} = \frac{1}{N_o} \ \forall i = 1, ..., N_o \right\}$$

that solves

$$P^\star = \arg \min_{P \in \mathcal{B}_o} \langle P, C \rangle + \rho \, KL \left( P^T \mathbf{1}_{N_o} \| \frac{\mathbf{1}_{N_s}}{N_s} \right) + \gamma \langle P, \log P \rangle, \tag{2}$$

where $\mathbf{1}_n$ is a vector of ones with size $n$ and $KL\,(.)$ is the Kullback-Leibler divergence between the marginal distribution over the simulated observations implied by the transport matrix $P$ and the uniform distribution. Therefore, a larger $\rho > 0$ promotes a coupling that fits the simulated data more closely, and $\gamma > 0$ is a hyperparameter that encourages entropic transport matrices. This problem can be solved with a variant of the Sinkhorn algorithm (Cuturi, 2013) with efficient GPU implementations. In our experiments, we rely on OTT (Cuturi et al., 2022) to return such a coupling $P^\star$, given $C$, the entropic regularization factor $\gamma$, and $\rho$, parameterized as $\tau = \rho/\rho + \gamma$. Setting $\tau = 1$ recovers a perfectly balanced transport.

## 3 MODELING MISSPECIFICATION WITH OT

In this section, we formally introduce our robust posterior estimation algorithm (RoPE) and highlight some benefits of modeling misspecification with OT. RoPE approaches the problem of misspecification as a hybrid modeling task by combining the simulator with a misspecification model learned from the few observations in the calibration set. The main modeling assumption of RoPE is

$$\mathbf{x}_o \perp \theta \mid \mathbf{x}_s, \tag{3}$$

which says that given the simulated observations $\mathbf{x}_s$, the real observations $\mathbf{x}_o$ contain no additional information about the parameters $\theta$. As a consequence, we can express the posterior for real-world observations as $p(\theta \mid \mathbf{x}_o) = \int p(\theta \mid \mathbf{x}_s) p(\mathbf{x}_s \mid \mathbf{x}_o) \mathrm{d}\mathbf{x}_s$, where $p(\theta \mid \mathbf{x}_s)$ is easily approximated with NPE. On the other hand, the conditional $p(\mathbf{x}_s \mid \mathbf{x}_o)$, which can be attributed to misspecification is what RoPE intends to learn by estimating an OT coupling (that is then conditioned on $\mathbf{x}_0$).

This assumption does not prevent obtaining calibrated and informative posterior distributions, even when it does not hold. Moreover, the assumption acts as a regularizer that allows learning a generalizable misspecification model from only a tiny calibration set. It also ensures predictions follow from the expert knowledge embedded in the simulator. This information bottleneck is a limiting factor for highly misspecified simulators that poorly model the dependencies between parameters and observations. However, suppose the simulator encodes phenomena the practitioner believes are invariant across different application environments; then, the assumption also prevents shortcut learning from the calibration data and benefits the generalization of the method. In Appendix D, we evaluate the method on real out-of-distribution data and demonstrate this property.

Intuitively, the OT coupling obtained from solving (2) defines a joint distribution $\pi^\star$ in $\mathcal{X}_o \times \mathcal{X}_s$ when $\tau = 1$ (see Appendix E for further discussion). Thus, together with our modeling assumption (3), we can express the posterior distribution for real-world observations as

$$p(\theta \mid \mathbf{x}_o) = \int p(\theta \mid \mathbf{x}_s) \pi^\star(\mathbf{x}_s \mid \mathbf{x}_o) \mathrm{d}\mathbf{x}_s, \tag{4}$$

where the simulation posterior $p(\theta \mid \mathbf{x}_s)$ can be approximated with NPE (Papamakarios & Murray, 2016), as NFs are universal density estimators of continuous distributions (Wehenkel & Louppe, 2019; Draxler et al., 2024).

Motivated by the factorization in (4), our algorithm computes a transport matrix $P^\star$ between the test set $\mathcal{D}$ and a set $\{\mathbf{x}_s^j\}_{j=1}^{N_s}$ of $N_s$ simulations generated by running the simulator on parameters from the given prior $\theta^j \sim p(\theta)$. Thus, approximating (4), we estimate the posterior for real-world observations as a mixture of posteriors $\tilde{p}$ obtained with NPE, that is,

$$\tilde{p}(\theta \mid \mathbf{x}_o^i) := \sum_{j=1}^{N_s} \alpha_{ij} \tilde{p}(\theta \mid \mathbf{x}_s^j), \text{ where } \alpha_{ij} = N_o P_{ij}^\star. \tag{5}$$

## 3.1 DEFINING THE OT COST FUNCTION

In our context, an ideal coupling would assign simulations to real-world observations generated from the same parameter. Hence, we can express the corresponding ideal cost as $c(\mathbf{x}_o, \mathbf{x}_s) = c(\mathbf{h}_o(\mathbf{x}_o), \mathbf{h}_s(\mathbf{x}_s))$, where $\mathbf{h}_o$ and $\mathbf{h}_s$ are any sufficient statistics for $\theta$ given $\mathbf{x}_o$ and $\mathbf{x}_s$, respectively.

As discussed in Appendix G, we can learn an approximated minimal sufficient statistic $\mathbf{h}_{\omega^\star}$ for the simulated observations with NPE. Furthermore, as the simulator carries information about the true generative process and the calibration set is too small to learn a representation only from real-world data, it is reasonable to learn a sufficient statistic $\mathbf{h}_o$ for the real observations by fine-tuning $\mathbf{h}_{\omega^\star}$. Denoting this new neural network as $\mathbf{g}_\varphi : \mathcal{X}_o \to \mathbb{R}^l$, the fine-tuning objective reads

$$\mathcal{L}(\varphi; \mathcal{C}) := \sum_{i=1}^{N_c} |\mathbf{g}_\varphi(\mathbf{x}_o^i) - \mathbb{E}_{\varepsilon \sim \mathcal{U}[0,1]}[\mathbf{h}_{\omega^\star}\left(S(\theta^i, \varepsilon)\right)]|_2, \tag{6}$$

where the expectation is approximated via a Monte-Carlo approximation. The training of $\mathbf{g}$ starts from the weights $\omega^\star$ and optimizes (6) with gradient descent. Optimizing (6) enforces, at least on the calibration set, that $\mathbf{g}$ and $\mathbf{h}$ are close in L2 norm when they correspond to the same parameter. Thus, we define the OT cost as $c(\mathbf{x}_o, \mathbf{x}_s) := |\mathbf{g}_{\varphi^\star}(\mathbf{x}_o) - \mathbf{h}_{\omega^\star}(\mathbf{x}_s)|_2$, where $\mathbf{g}_{\varphi^\star}$ is the NSE obtained after fine-tuning (6). Figure 1 depicts the main training and inference steps of RoPE. We discuss the computational cost of RoPE in Appendix H.

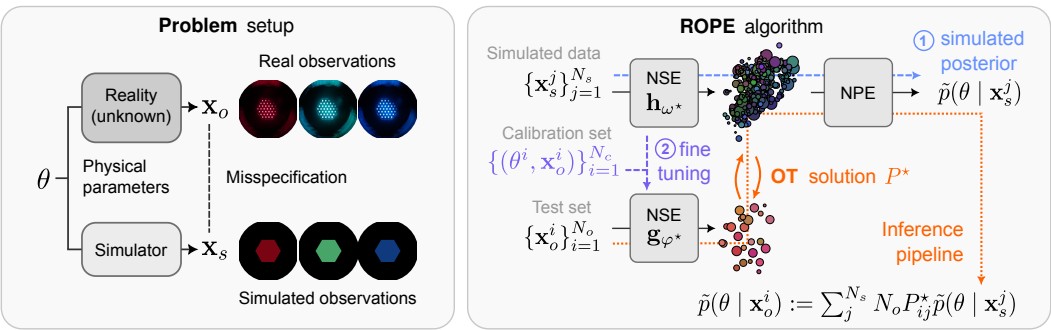

Figure 1: (*left*) Problem setup: we consider a real-world process which depends on some physical parameters $\theta$. Given real observations $\mathbf{x}_o$ of the process, our goal is to provide uncertainty quantification on the underlying parameters $\theta$. To help us, we have access to a misspecified simulator that takes parameters $\theta$ as input and produces simulated observations $\mathbf{x}_s$. (*right*) A visualization of RoPE. The training consists of two steps: (1) given the simulated data, we approximate the posterior using NPE, resulting in the NSE $\mathbf{h}_{\omega^\star}$; (2) using the calibration set, we fine-tune $\mathbf{h}_{\omega^\star}$ into $\mathbf{g}_{\varphi^\star}$ using the objective (6). At test time, we solve the optimal transport (OT) problem between the representations $\{\mathbf{h}_{\omega^\star}(\mathbf{x}_s^j)\}_{j=1}^{N_s}$ and $\{\mathbf{g}_{\varphi^\star}(\mathbf{x}_o^i)\}_{i=1}^{N_o}$, resulting in our estimated posterior (5), the average of simulations' posteriors weighted by the OT solution $P^\star$. See Algorithm 1 in Appendix B for more details.

## 3.2 ON THE BENEFITS OF USING OPTIMAL TRANSPORT TO HANDLE MISSPECIFICATION

Several attractive properties of RoPE directly follows from modeling the misspecification as an OT coupling between simulated and real-world measurements. First, **a self-calibration property**: by modeling the posterior as (5), when $\tau = 1$ (i.e., the transport is perfectly balanced), the marginal posterior distribution over the test set, i.e., $\tilde{p}(\theta) := \int \tilde{p}(\theta \mid \mathbf{x}_o) p^\star(\mathbf{x}_o) d\mathbf{x}_o$, converges to the prior distribution as the number of simulated observations $N_s$ approaches infinity, as expected from a well-estimated posterior distribution. A proof and further discussion of this self-calibration property is given in Appendix F. Second, **a control mechanism for the posteriors' confidence**: the entropic regularization of OT not only enables fast computation of the transport coupling but also provides an effective control mechanism to balance the calibration of the posterior with its informativeness. Indeed, for small entropic regularization, the estimated posteriors have low entropy and may be overconfident, as they are sparse mixtures of a few simulation posteriors $\tilde{p}(\theta \mid \mathbf{x}_s^j)$. In contrast, for large values of $\gamma$ in (2), the coupling matrix becomes uniform and the corresponding posteriors tend to the prior, as $p(\theta \mid \mathbf{x}_o) \approx \frac{1}{N_s} \sum_j^{N_s} \tilde{p}(\theta \mid \mathbf{x}_s^j)$ is a Monte-Carlo approximation of $\mathbb{E}_{p(\mathbf{x}_s)}[\tilde{p}(\theta \mid \mathbf{x}_s)] \approx p(\theta)$. Thus, the practitioner should optimize the hyper-parameter $\gamma$ to find the right trade-off between calibration of the estimated posteriors, favored by higher $\gamma$, and their informativeness, favored by

lower $\gamma$ (see Figure 3). Finally, **robustness to prior misspecification**: by enabling the transport to be unbalanced—that is, to discard simulated observations when $\tau < 1$—RoPE can flexibly depart from the assumed marginal distribution of $p(\theta)$ and be robust to prior misspecification (Figure 4). Thus, the parameter $\tau$ can be seen as a control mechanism to account for the user's confidence in the prior distribution. In the rest of the text, we denote the method as RoPE$^\star$ when $\tau < 1$ and as RoPE when $\tau = 1$. In subsection 4.1, we provide guidance on how to set $\gamma$ and $\tau$ in practice.

# 4 EXPERIMENTS

Our experiments aim to (1) empirically validate the discussion in Section 3.2, and (2) illustrate settings in which our algorithm enables uncertainty quantification under model misspecification and small calibration datasets. The experiments comprise two existing benchmarks from the SBI literature, two synthetic benchmarks, and two new benchmarks from real physical systems for which both labeled data and simulators are available. To the best of our knowledge, the latter constitute the first real-world benchmarks for SBI under misspecified models that directly provide a ground truth for the underlying parameters $\theta$. Altogether, the benchmarks represent various types of misspecification and parameter and observation space. We briefly describe each task and provide examples of real vs. simulated observations in Figure 2. Further details about the experiments can be found in Appendix I.

**Task A & B (synthetic): CS & SIR .** We reproduce the cancer and stromal cell development (CS) and the stochastic epidemic model (SIR) benchmarks from Ward et al. (2022). We provide a description of the parameters, observations and synthetic misspecification in subsection I.1

**Task C (synthetic): Pendulum.** The damped pendulum is a common benchmark to assess hybrid learning algorithms (Wehenkel et al., 2022), which jointly exploit domain knowledge and real-world data. The simulator generates the horizontal position of a friction-less pendulum given its fundamental frequency $\omega_0 \in \mathbb{R}^+$ and amplitude $A \in \mathbb{R}^+$. Randomness enters the simulator through a random phase shift and white measurement noise. As misspecified "real-world" data, we simulate observations from a damped pendulum that takes friction into account.

**Task D (synthetic): Hemodynamics.** Following Wehenkel et al. (2023), we define the task of inferring the stroke volume (SV) and the left ventricular ejection time (LVET) from normalized arterial pressure waveforms. The simulator is a PDE solver (Melis, 2017) that produces an 8-second time-series $\mathbf{x}_s$ sampled at 125Hz. As synthetic misspecification, the simulator assumes all arteries have constant length, whereas this parameter varies in the "real-world" data.

**Task E (real): Light Tunnel.** We employ one of the light tunnel datasets from Gamella et al. (2024). The tunnel is an elongated chamber with a controllable light source at one end, two linear polarizers mounted on rotating frames, and a camera. Our task consists of predicting the color setting of the light source ($(R, G, B) \in [0, 255]^3$) and the dimming effect of the polarizers $\alpha \in [0, 1]$ from the captured images. The simulator takes the parameters $\theta := [R, G, B, \alpha]$ and produces an image consisting of a hexagon roughly the size of the light source, with a color equal to $[\alpha R, \alpha G, \alpha B]$.

**Task F (real): Wind Tunnel.** We employ one of the wind tunnel datasets from Gamella et al. (2024). The tunnel is a chamber with two controllable fans that push air through it, and barometers that measure air pressure at different locations. A hatch controls the area of an additional opening to the outside. The dataset is a collection of pressure curves that result from applying a short impulse to the intake fan power and measuring the change in air pressure inside the tunnel. Our inference task consists of predicting the hatch position, $\theta := H \in [0, 45]$ given a pressure curve. As a simulator model, we adapt the physical model given in Gamella et al. (2024, Appendix IV).

**Metrics.** We consider two metrics to assess whether RoPE provides reliable and useful uncertainty quantification. First, given a labeled test set $\{(\theta^i, \mathbf{x}_o^i)\}_{i=1}^N$, we compute the log-posterior probability (LPP) as LPP $:= \frac{1}{N} \sum_{i=1}^N \log \tilde{p}(\theta^i \mid \mathbf{x}_o^i) \approx \mathbb{E}_{\substack{p(\mathbf{x}_o) \\ p(\theta \mid \mathbf{x}_o)}} [\log \tilde{p}(\theta \mid \mathbf{x}_o)]$. The LPP is an empirical estimation of the expectation over possible observations of the negative cross entropy between the true and estimated posterior; thus, for an infinite test set, it is only maximized by the true posterior. LPP characterizes the entropy reduction on the estimation of $\theta$ achieved by a posterior estimator $\tilde{p}$ when given one observation, on average, over the test set. Second, the average coverage AUC (ACAUC) indicates the average calibration of $K$ 1D credible intervals extracted from the estimated posteriors, i.e., ACAUC $:= \frac{1}{KN} \sum_{j=1}^K \sum_{i=1}^N \int_0^1 \alpha - \mathbf{1}[\theta_j^i \in \Theta_{\tilde{p}(\theta_j \mid \mathbf{x}_o^i)}(\alpha)] \mathrm{d}\alpha$, where $\Theta_{\tilde{p}(\theta_j \mid \mathbf{x}_o^i)}(\alpha)$ denotes the

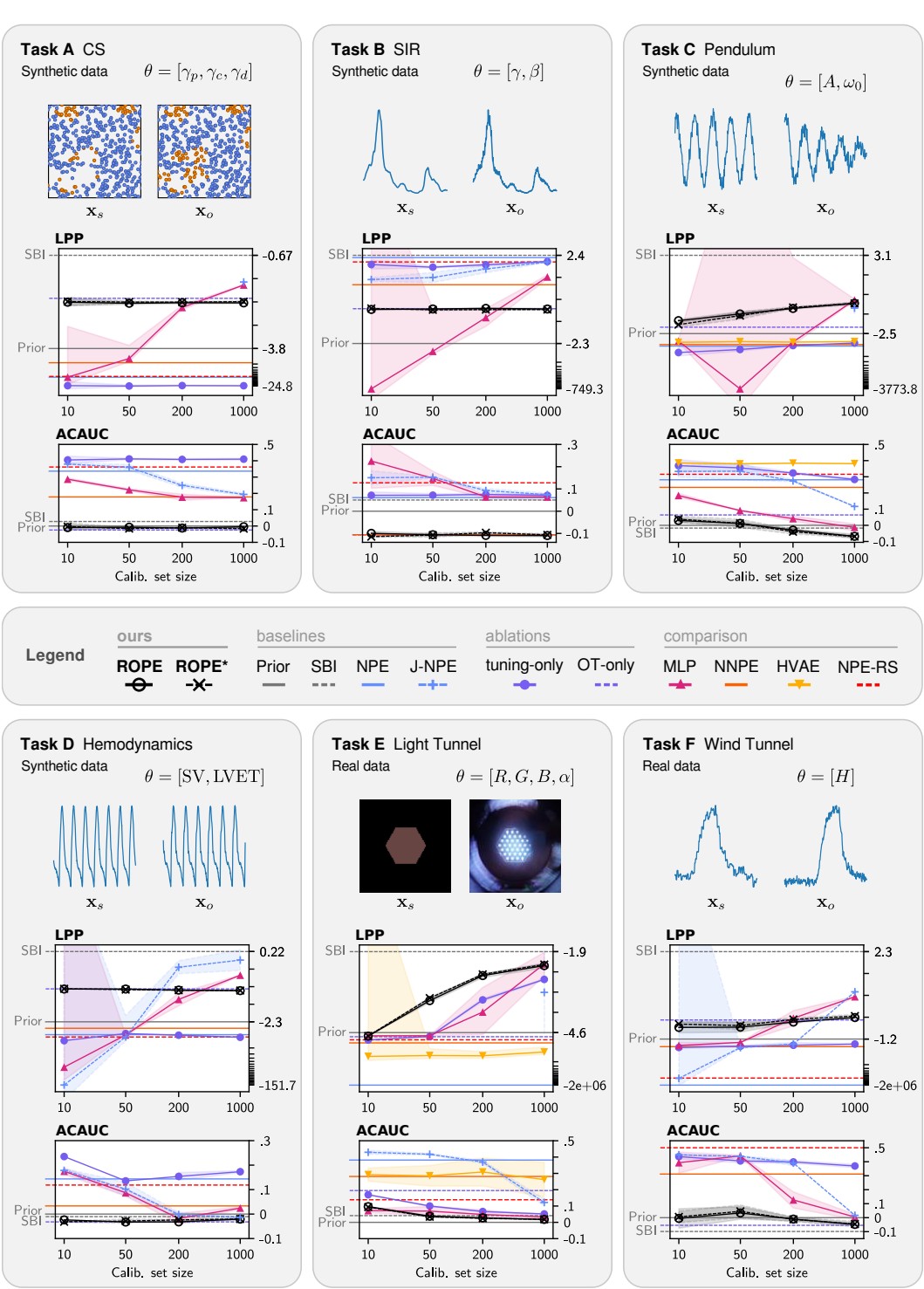

Figure 2: Results for our method (RoPE) and the competing baselines on six benchmark tasks. For each task, we show an example of the real observations ($\mathbf{x}_o$) and the observations produced by the misspecified simulator ($\mathbf{x}_s$). We show each method's LPP and ACAUC metrics, as computed on a labeled test set of size 2000. Horizontal lines without markers correspond to the methods that do not use the calibration set, producing a constant score. We report the average metrics and $\pm 1$ std. deviation over three random draws of the test set and additional sources of randomness. In some instances, e.g., NPE-RS in task C, the likelihood can be $-\infty$ and is not plotted. For readability of the LPP metric, we use a linear scale between the SBI and the Prior and a logarithmic scale for values below that.

credible interval for the j[th] dimension of the parameter $\theta$ at level $\alpha$. Its value is positive (negative) if, on average over different credible levels, parameter dimensionality, and observations, the corresponding credible intervals are overconfident (underconfident). The ACAUC of a perfectly specified prior distribution is zero. The integral can be efficiently approximated, as described in Appendix J. For all experiments, we compute the LPP and ACAUC on labeled test set containing 2000 pairs $(\theta, \mathbf{x}_o)$.

**Baselines.** As a sanity check, we compare the performance of RoPE against four reference baselines: the **prior** $p(\theta)$, which amounts to the lower bound on the LPP for any calibrated posterior estimator when the prior is well-specified; the **SBI** posterior, which is an NPE trained and tested on simulated data and thus provides an upper bound on the LPP for RoPE under the independence assumption $\mathbf{x}_o \perp \theta \mid \mathbf{x}_s$ (see Appendix I for more details); (**NPE**) a posterior estimator fitted to the simulated data and applied to the real data; and (**J-NPE**) a posterior estimator trained jointly on the pooled simulated and real observations. The latter two baselines represent some first approaches that a practitioner may consider. Furthermore, to asses how a fully supervised approach would fare if trained directly on the calibration set, we compare the performance of RoPE to **MLP**, which trains a neural network to predict the mean and log-variance of a Gaussian posterior distribution by maximizing the calibration set log-likelihood. We train both the MLP and J-NPE baselines in a supervised way, and we thus expect these baselines to perform strongly as the size of the calibration becomes sufficiently large, when the test data is i.i.d. We also run **NPE-RS** (Huang et al., 2023), which trains a robust version of NPE with a regularization loss that enforces the distributions of NSE on simulated and test data to match. For a fair comparison with RoPE, we use the $N = 2000$ test examples to compute the regularization, informing NPE-RS as much as possible. We additionally run Noisy NPE (**NNPE**, Ward et al., 2022), the amortized version of RNPE introduced in the same paper, which improves the robustness of NPE by introducing a Spike and Slab error model on simulated data statistics. We also run **HVAE** (Takeishi & Kalousis, 2021), which constitutes a strong baseline when the simulator can be made differentiable (tasks C and E) but is not directly applicable otherwise. More details about each method and the experimental setup can be found in Appendix I.

### 4.1 RESULTS

In Figure 2, we compare the performance of RoPE against the baselines and other methods for the six tasks we consider with a correctly specified prior. To demonstrate that applying RoPE is straightforward, we deliberately fix $\gamma = 0.5$ for RoPE and $\tau = 0.9$ for RoPE$^\star$ in all six tasks. In Figure 3 and Figure 4, we study the role of RoPE's hyperparameters, which can be further tuned to optimize performance if the practitioner can relate the estimated posterior to a validation metric of interest.

**RoPE achieves robust posterior estimation for all tasks.** As mentioned above, the SBI and prior baselines provide upper and lower bounds on the expected performance of a well-calibrated posterior estimator, under the modeling assumption made in section 3. For all tasks, even with minimal calibration budgets, RoPE is the only method that consistently returns well-calibrated, or sometimes slightly under-confident, posterior estimation while significantly reducing uncertainty compared to the prior distribution. As the size of the calibration set increases, we see the adaptability of J-NPE and MLP as their performance improves and aligns with or outperforms RoPE. This adaptability is an expected behavior in i.i.d. settings, where real-world data eventually allows finding the minimizer of empirical risk among a class of predictors. Nevertheless, these two baselines tend to be overconfident even for larger calibration sets, as highlighted by their positive ACAUC numbers, which are significantly larger than RoPE's ACAUC in almost all configurations. Moreover, on task E, where posteriors are complex conditional distributions—whose entropy increases with darker images and contain non-trivial dependencies between parameters—RoPE remains the best approach, even with a calibration set containing more than 1000 examples. As an outlier, we observe that NPE trained on simulated data achieves the best results for the SIR benchmark (Task B), indicating that the misspecification of this benchmark is not a challenging test case for existing SBI methods and may thus not be ideal to benchmark methods that cope with model misspecification. Finally, because interpreting a numerical gap in LPP metrics can be difficult, we complement these numerical results with corner plots for the two real-world experiments in Figure 3 and for all tasks in Appendix K.1.

**Ablation study.** Our algorithm combines two steps with distinct roles: (1) a fine-tuning step, which improves the domain generalization of the NSE; and (2) an OT step, aiming to model the misspecification as a stochastic mapping between simulations and observations. To better understand

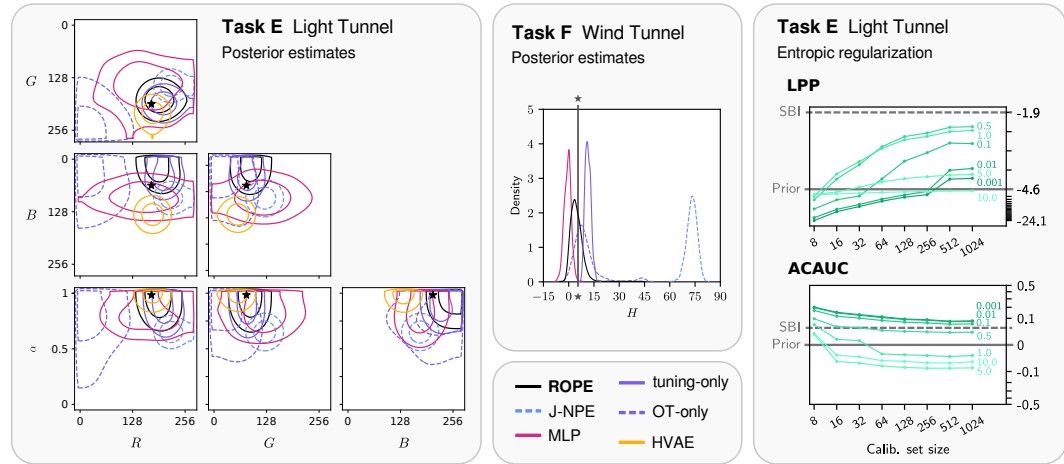

Figure 3: (*left*) Credible intervals of the posterior estimates at levels $65\%$ and $90\%$, for a single test sample from the light-tunnel task. The black stars denote the true value of the parameter. (center) Posterior estimates for a single test sample from the wind-tunnel task, where the true parameter is denoted by a vertical black line. (*right*) **Effect of** $\gamma$ on the LPP and ACAUC scores of RoPE on the light-tunnel task for different sizes of the calibration set. The value of $\gamma$ is shown by each curve. For reference, we plot the metrics achieved by the SBI posterior and prior distribution on simulated data.

their respective contribution to the performance of RoPE, we look at two ablated versions of our algorithm: **tuning-only** which appends the fine-tuned NSE to the NF trained on simulated data $p_{\phi^\star}$ and directly applies it to the real observations without an OT step; and **OT-only**, which directly performs OT with L2-norm in the original NSE space $c(\mathbf{x}_o, \mathbf{x}_s) = |\mathbf{h}_{\omega^\star}(\mathbf{x}_o) - \mathbf{h}_{\omega^\star}(\mathbf{x}_s)|_2$. In Figure 2, we observe that tuning-only's results are poor except for Task B, where misspecification is negligible. In contrast, for tasks A, D, and F, OT-only exhibits performance on par with RoPE. Nevertheless, RoPE can significantly outperform OT-only, such as in tasks C and E where the misspecification is significant. We conclude that the OT step is crucial and fine-tuning is sometimes necessary—we recommend that practitioners first evaluate OT-only's performance and optimize the value of $\gamma$ before using a subset of the real-world data for fine-tuning.

**Effect of entropic regularization—setting** $\gamma$**.** In Figure 3, we study the effect of entropic regularization by varying the regularization parameter $\gamma$. For all values of $\gamma$, excluding $\gamma \geq 5$, we observe that both LPP and calibration consistently improve with the calibration set size. For large values of $\gamma$, the entropic regularization dominates and pushes toward a uniform mapping, resulting in posteriors that approximate the prior distribution and are barely affected by the calibration set size. These empirical results are consistent with the theoretical discussion in Subsection 3.2. As a recommendation for practitioners, our empirical evaluation suggests that values between $0.1$ and $1$ provide well-calibrated and precise credible intervals. Ideally, the practitioner shall keep a significant portion of the calibration set for validation, using it to optimize $\gamma$ based on the metrics of interest. If this is not possible, we recommend employing $\gamma = 0.5$, which offers sharp and calibrated posteriors on all our benchmarks.

**RoPE$^\star$ for prior misspecification—setting** $\tau$**.** In Figure 4, we consider two experiments to study the impact of prior misspecification on RoPE and its unbalanced version RoPE$^\star$. More details about the experimental setup can be found in Appendix C. The left panel in Figure 4 compares the performance of RoPE ($\gamma = 0.5$ and $\tau = 1$) and RoPE$^\star$ ($\gamma = 0.5$ and $\tau \in \{0.5, 0.9\}$) on an extension of Task E, where the ground-truth parameters of the test dataset come instead from a beta-binomial distribution, meaning the original prior, a uniform distribution, is misspecified. We first observe that RoPE's performance resists the prior misspecification; it provides well-calibrated and informative posteriors, as is visible in the corner plots of Figure 5 in Appendix C. While the gap between RoPE and RoPE$^\star$ is negligible in the case of a well-specified prior (see Task E in Figure 2), under prior misspecification RoPE$^\star$ leverages the additional flexibility in the OT solution and discards some of the simulated observations, achieving higher LPP. In the right panel of Figure 4, we extend task C to further investigate the impact of an increased prior misspecification and the role of $\tau$ to address it. As expected, when there is no prior misspecification RoPE (i.e, $\tau = 1$) achieves the best performance. As prior misspecification increases, using lower values of $\tau$ becomes preferable. From

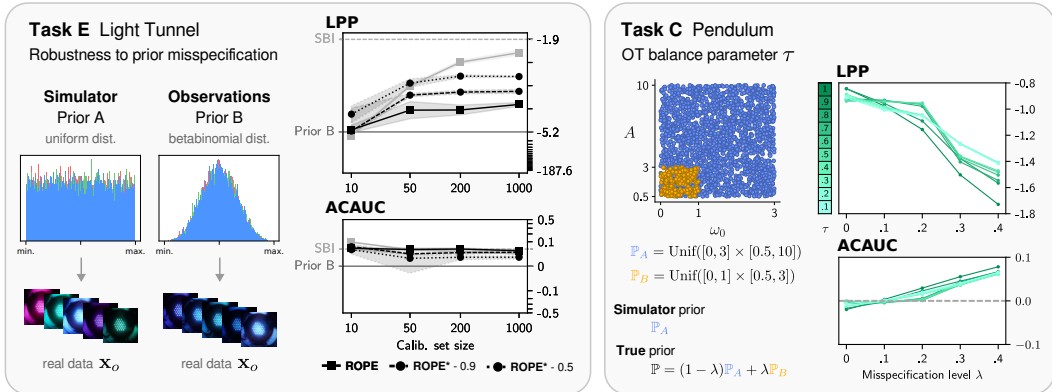

Figure 4: **Prior misspecification.** Evaluating the performance of RoPE when the prior used to generate the synthetic observations is incorrectly specified. *(left)* We report the performance of RoPE and RoPE$^\star$ (with $\tau = 0.9$ and $\tau = 0.5$), when tested on 2000 observations generated by sampling parameters from prior B, while the prior used to create simulations is prior A. For context, we also overlay the performance of RoPE on the Prior A in light gray. *(right)* We study the effect of $\tau \in [0.1, 1]$ under various levels of prior misspecification in the Pendulum experiments (task C). See Appendix C for further motivation and experimental details.

these experiments, we recommend leveraging $\tau$ as a hyperparameter describing confidence in the assumed prior distribution—setting its value to $0.9$ offers robust performance for both well-specified and partially misspecified priors. The user shall also explore lower values when there is suspicion that the prior distribution is overly spread with respect to the correct prior.

## 5 DISCUSSION

While our experiments demonstrate that RoPE efficiently leverages misspecified simulators and real-world data, its shortcomings open opportunities for future work, which we discuss here.

**Curse of dimensionality.** The dimensionality of $\theta$ may impact two critical parts of RoPE. First, with each additional parameter $\theta_{K+1}$, given $\mathbf{x}_o$, the NSE must encode up to $K$ dependencies between $\theta_{K+1}$ and the other dimensions $\theta_1, \ldots, \theta_K$. While generating more simulations can address the curse of dimensionality in the simulation space, fine-tuning on a small calibration may no longer suffice to cope with misspecification. Second, the dimensionality of the manifold on which the NSE projects the simulated and real-world observations will grow, and finding a meaningful coupling between the two populations may require larger sample sizes. A potential solution is to focus marginal or 2D posterior distributions and ignore higher-dimensional dependencies in $p(\theta \mid \mathbf{x}_o)$. Nevertheless, extending RoPE to such settings certainly opens new questions, e.g., concerning the development of better fine-tuning strategies that can leverage partial calibration sets where labels can be incomplete.

**Other extensions.** Similar to incomplete labels, in certain applications we may only have access to noisy labels, measured with a well-modeled but noisy measurement process. Further developing the fine-tuning stage to exploit such noisy labels would be necessary to make an approach similar to RoPE applicable. As another direction, leveraging inductive bias embedded into the neural network architecture of neural OT, the ability to better cope with a large test set appears as a promising direction to amortize the mapping between simulation and real-world data. We believe following RoPE's strategy of modeling misspecification in SBI as an OT coupling opens up several avenues to address more specific problem setups.

**Conclusion.** In this paper, we show that model misspecification in simulation-based inference can be addressed using a small calibration set of labeled real-world data. We have argued that there are important settings where such calibration sets are the norm but where SBI is not applied due to its sensitivity to model misspecification. Under this premise, we have introduced RoPE, an algorithm that jointly exploits a small calibration set and optimal transport to extend neural posterior estimation for misspecified simulators. Our experiments on diverse benchmarks demonstrate RoPE's ability to estimate well-calibrated and informative posterior distributions for various simulators and real-world examples. In conclusion, RoPE is a simple, yet flexible and effective, method that allow practitioners to predict a calibrated posterior over the parameters of a misspecified simulator from real-world data.

**Ethics Statement** This paper presents a framework and an algorithm to address model misspecification in simulation-based inference (SBI). SBI is predominantly applied in scientific fields where complex simulators of physical phenomena are available, such as astronomy, medicine, particle physics, or climate modeling. A priori, this circumscribes the application of our algorithm to highly specialized scientific domains in the natural sciences, precluding issues such as fairness or privacy. However, its application to the scientific domain is not exempt from societal or ethical implications, particularly when computer simulations may inform research or policy decisions. In this regard, we find some properties of the algorithm particularly promising, such as uncertainty quantification and the limitation of not drawing conclusions beyond the given expert model. However, more work is needed to deeply understand the reliability of these properties. Such work should precede any sort of over-selling to practitioners about the benefits of the algorithm. Rather, we see our work as a contribution towards a more broad and successful application of SBI techniques; success in this endeavor, as for the establishment of any scientific tool, will require an iterative dialogue between the scientists who develop the methodology and those who use it.

**Reproducibility Statement** We will provide the accompanying code for reproducing all the results with the camera-ready version of the manuscript. Nevertheless, we already provide a thorough description of the experimental setup in Appendix I together with links to the datasets. We also provide a rigorous description of our algorithm, including the toolbox used to solve the OT problem, in the main text and Appendix B.

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
