# A  MODEL MISSPECIFICATION

## A.1  MIS-CALIBRATION VS MISSPECIFICATION

To further elucidate the distinction between posterior calibration and model misspecification, it is essential to highlight their respective scopes and the specific challenges they address.

Posterior calibration focuses on ensuring that the predicted posterior distributions accurately reflect the true uncertainty in parameter estimates given the observations, under the assumption that the simulator is well-specified. Methods such as those proposed by Falkiewicz et al. (2024); Delaunoy et al. (2022) address this by improving the alignment between the expected and actual coverage probabilities of the posterior. These approaches generally assume that the simulator faithfully represents the generative process of the observed data, enabling calibration to be evaluated and improved by leveraging simulations. While important, these methods do not account for discrepancies between the simulator and real-world data, which are precisely the scenarios we target in this work.

Model misspecification, on the other hand, arises when the simulator fails to capture the true generative process underlying the observed data. This results in systematic discrepancies that cannot be corrected solely by optimizing posterior calibration techniques. Misspecification introduces a gap between the simulated and real-world distributions, and this gap is only observable when real-world data is available. Unlike posterior calibration, addressing misspecification requires methods that can robustly leverage the simulator despite its inaccuracies, while incorporating real-world observations to mitigate the impact of the mismatch.

In our work, we explicitly focus on handling model misspecification. This distinction is reflected in the design of our approach and the evaluation scenarios we consider, such as Task E, where the simulated data diverges significantly from the real-world measurements. While posterior calibration methods may perform well in a well-specified context, they are not designed to cope with such gaps. Instead, we prioritize creating predictive models that balance informativeness and robustness in the presence of misspecification, even if achieving perfect calibration remains an open and challenging problem.

## A.2  COMPARISON BETWEEN MODEL MISSPECIFICATION DEFINITIONS

We provide a toy example to show how a simulator may be well-specified according to the standard definition of misspecification but still provide biased estimates of the target parameter when applied to real data.

Consider the following setting: a noisy sensor measures some physical quantity $\theta$, producing measurements $\mathbf{x}_o^1, \ldots, \mathbf{x}_o^n \overset{\text{i.i.d.}}{\sim} \mathbb{P}^\star$, where $\mathbb{P}^\star := \mathcal{N}(\theta^\star, 1)$ is a normal distribution centered around the 'true' value $\theta^\star$. Let $\{\mathbb{P}_\theta : \theta \in \mathbb{R}\}$ be a simulator of this process with $\mathbb{P}_\theta := \mathcal{N}(\mu, 1)$, where $\mu := \theta + \lambda$ and $\lambda > 0$ is a fixed scalar constant, which is a misspecification in the simulator that falsely accounts for a non-existing offset in the sensor that produced the real observations $\mathbf{x}_o^1, \ldots, \mathbf{x}_o^n$.

According to the standard definition of misspecification, the simulator is well specified, as setting $\theta \leftarrow \theta^\star - \lambda$ yields $\mathbb{P}_\theta = \mathbb{P}^\star$. However, the posterior estimates we obtain with this simulator are biased with respect to the true parameter $\theta^\star$.

To see this, let us compute the posterior under a Gaussian prior $\mathcal{N}(\theta^\star, 1)$ over the parameter $\theta$, centered on the true value $\theta^\star$. Taking advantage of the conjugate prior, the posterior $p(\theta \mid \mathbf{x}_o^1, \ldots, \mathbf{x}_o^n)$

becomes

$$p(\theta \mid \mathbf{x}_o^1, \ldots, \mathbf{x}_o^n) \propto p(\theta)p(\mathbf{x}_o^1, \ldots, \mathbf{x}_o^n \mid \theta)$$

$$= p(\theta) \prod_{i=1}^n p(\mathbf{x}_o^i \mid \theta)$$

$$= \frac{1}{\sqrt{2\pi}} \exp\left(-\frac{1}{2}(\theta - \theta^\star)^2\right) \prod_{i=1}^n \frac{1}{\sqrt{2\pi}} \exp\left(-\frac{1}{2}(\mathbf{x}_o^i - \mu)^2\right)$$

$$\propto \exp\left(-\frac{1}{2}(\theta - \theta^\star)^2 - \frac{1}{2}\sum_{i=1}^n (\mathbf{x}_o^i - \mu)^2\right)$$

$$= \exp\left(-\frac{1}{2}\left[\theta^2 + (\theta^\star)^2 - 2\theta\theta^\star + \sum_{i=1}^n (\mathbf{x}_o^i)^2 + n\mu^2 - 2\mu\sum_{i=1}^n \mathbf{x}_o^i\right]\right)$$

$$\text{(drop const. terms)} \quad \propto \exp\left(-\frac{1}{2}\left[\theta^2 - 2\theta\theta^\star + n\mu^2 - 2\mu\sum_{i=1}^n \mathbf{x}_o^i\right]\right)$$

$$(\mu = \theta + \lambda) \quad = \exp\left(-\frac{1}{2}\left[\theta^2 - 2\theta\theta^\star + n\theta^2 + n\lambda^2 + 2n\lambda\theta - 2\theta\sum_{i=1}^n \mathbf{x}_o^i - 2\lambda\sum_{i=1}^n \mathbf{x}_o^i\right]\right)$$

$$\text{(drop const. terms)} \quad \propto \exp\left(-\frac{1}{2}\left[\theta^2 - 2\theta\theta^\star + n\theta^2 + 2n\lambda\theta - 2\theta\sum_{i=1}^n \mathbf{x}_o^i\right]\right)$$

$$= \exp\left(-\frac{1}{2}\left[(n+1)\theta^2 - 2\theta(\theta^\star - n\lambda + \sum_{i=1}^n \mathbf{x}_o^i)\right]\right)$$

$$= \exp\left(-\frac{1}{2(n+1)^{-1}}\left[\theta^2 - 2\theta\left(\frac{1}{n+1}\right)(\theta^\star - n\lambda + \sum_{i=1}^n \mathbf{x}_o^i)\right]\right)$$

$$\text{(complete square)} \quad \propto \exp\left(-\frac{1}{2(n+1)^{-1}}\left[\theta - \left(\frac{1}{n+1}\right)(\theta^\star - n\lambda + \sum_{i=1}^n \mathbf{x}_o^i)\right]^2\right),$$

that is, a normal distribution $\mathcal{N}(\tau, \gamma^2)$ with mean

$$\tau = \left(\frac{1}{1+n}\right)\left(\theta^\star - n\lambda + \sum_{i=1}^n \mathbf{x}_o^i\right)$$

and variance $\gamma^2 = (n+1)^{-1}$. Thus, the posterior is biased, e.g., the posterior mean $\tau$ is a biased estimator of $\theta^\star$ with $\mathbb{E}[\theta^\star - \tau] = \theta^\star - \lambda\left(\frac{n}{n+1}\right)$.

# B  THE RoPE ALGORITHM

---

**Algorithm 1** Posterior Inference using Robust Neural Posterior Estimation (RoPE)

---

**Input:** Simulator $S(\theta, \varepsilon)$, prior distribution $p(\theta)$, calibration set $\mathcal{C} = \{(\mathbf{x}_o^i, \theta^i)\}_{i=1}^{N_c}$, test set $\mathcal{D} = \{\mathbf{x}_o^i\}_{i=1}^{N_o}$

**Output:** $\tilde{p}(\theta \mid \mathbf{x}_o) \forall \mathbf{x}_o^i \in \mathcal{D}$

**Step 1: Neural Posterior Estimation (NPE)**

Train neural network $\mathbf{h}_\omega$ and conditional normalizing flow $p(\theta \mid \cdot)$ using NPE:

$$\tilde{p}, \omega^\star = \arg\max_{p,\omega} \mathbb{E}_{\substack{\theta \sim \pi(\theta) \\ \varepsilon \sim \mathcal{U}[0,1]}} \left[ \log p(\theta \mid \mathbf{h}_\omega(S(\theta, \epsilon))) \right]$$

**Step 2: Fine-tune sufficient statistics $\mathbf{h}_{\omega^\star}$ on the Calibration Set**

$\mathbf{g}_\psi := \text{COPY}(\mathbf{h}_{\omega^\star})$

$\mathcal{C}_{train}, \mathcal{C}_{val} = \text{RandomSplit}(\mathcal{C}, \frac{1}{5})$

$best_{val} = \infty$

**for** $N_{\text{iter}}$ **do**

$\quad \psi \leftarrow \psi - \alpha \nabla_\psi \left[ \sum_{(\theta, \mathbf{x}_o) \in \mathcal{C}_{train}} |\mathbf{g}_\psi(\mathbf{x}_o) - \mathbb{E}_\varepsilon[\mathbf{h}_{\omega^\star}(S(\theta, \varepsilon))]|_2 \right]$

$\quad \text{cur}_{val} = \sum_{(\theta, \mathbf{x}_o) \in \mathcal{C}_{val}} |\mathbf{g}_\psi(\mathbf{x}_o) - \mathbb{E}_\varepsilon[\mathbf{h}_{\omega^\star}(S(\theta, \varepsilon))]|_2$

$\quad$ **if** $\text{cur}_{val} < best_{val}$ **then**

$\quad\quad best_{val} = \text{cur}_{val}$

$\quad\quad \psi^\star = \psi$

$\quad$ **end if**

**end for**

**Step 3: Generate Simulations for Test Set ($N_s = N_o$)**

$\mathcal{S} = \{\mathbf{x}_s^j\}_{j=1}^{N_s}$,

where $\mathbf{x}_s^j \sim S(\theta^j, \varepsilon) \quad \theta^j \sim \pi(\theta) \quad \varepsilon \sim \mathcal{U}[0,1]$

**Step 4: Entropic-regularized OT**

$$C_{ij} = |f_{\omega^\star}(\mathbf{x}_s^j) - g_{\psi^\star}(\mathbf{x}_o^i)| \quad \forall i, j \in \{1, \dots, N_o\} \times \{1, \dots, N_s\}$$

$$P^\star = \arg\min_{P \in \mathcal{B}_o} \langle P, C \rangle + \rho \, KL\left(P^T \mathbf{1}_{N_o} \Big\| \frac{\mathbf{1}_{N_s}}{N_s}\right) + \gamma \langle P, \log P \rangle$$

**Step 5: Compute Posterior Distributions**

$$p(\theta | \mathbf{x}_o^i) := \sum_{j=1}^{N_s} P_{ij}^\star \tilde{p}\left(\theta \mid \mathbf{h}_{\omega^\star}(\mathbf{x}_s^j)\right)$$

**Return** $\tilde{p}(\theta | \mathbf{x}_o^i) \quad \forall \mathbf{x}_o^i \in \mathcal{D}$

---

## C  ROBUSTNESS TO PRIOR MISSPECIFICATION

In some practical applications of our algorithm, it is unlikely that the prior used to generate synthetic data will match the distribution of the target parameters in the real data. For this reason, we consider a semi-balanced formulation of OT, providing the flexibility to discard simulations with no corresponding real-world observations.

**Prior misspecification on Task E.** To evaluate the effect of a misspecified prior on RoPE and RoPE⋆, we perform an experiment that would resemble its use in real applications like the ones we outline in the introduction. In such settings—e.g., inferring cardiac parameters or chemical concentrations—the target parameters are limited to a range of validity, and a likely choice for the practitioner would be to select a uniform prior over this range.

To replicate this setting, we collect a new real-world dataset from the light tunnel (Task E) and train RoPE on synthetic data originating from a uniform prior, as we do for the results shown in Figure 2. However, we then apply RoPE to real data generated from a different (betabinomial) distribution over the target parameters. The results are shown in Figure 4, together with a visualization of the misspecified and true parameter distributions (prior A and B, respectively). We also show the learnt posterior distributions in Figure 5 for both RoPE and RoPE⋆ ($\tau = 0.5$).

The results show that RoPE is relatively robust to prior misspecification prior. Furthermore, using an unbalanced OT formulation significantly improves the performance in this setting.

**Prior misspecification on Task C.** With this experiment we aim to better understand the role of $\tau$ when RoPE is applied with different levels of prior misspecification. We thus re-use the same setup as in Figure 2 but add prior misspecification as a mixture between the assumed prior and a much tighter uniform distribution. As the weight of the tighter uniform distribution increases, the prior gets more misspecified. The experimental setup follows closely the one in the well-specified case (see subsection I.2), except calibration samples are drawn from the true prior (as this would be the case in a real-world application) and we compute the OT coupling for values of $\tau \in [0.1, 1]$.

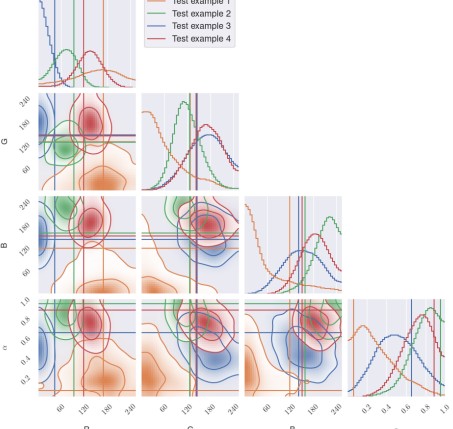 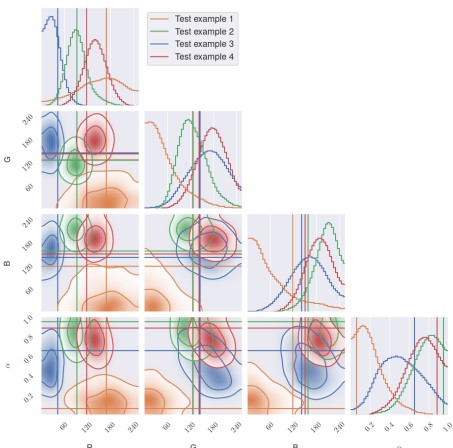

Figure 5: **Visualization of estimated posteriors.** Corner plots of the posteriors estimated by RoPE in the prior-misspecification experiment from Fig. 1 above. We show, in different colors, the estimates for four observations sampled at random from the test set, for RoPE (left) and RoPE⋆ ($\tau = 0.5$) (right) formulation of the OT step, and a calibration set of size 50; the horizontal and vertical lines correspond to the ground-truth value of the parameters.

# D    ROBUSTNESS TO DISTRIBUTION SHIFTS

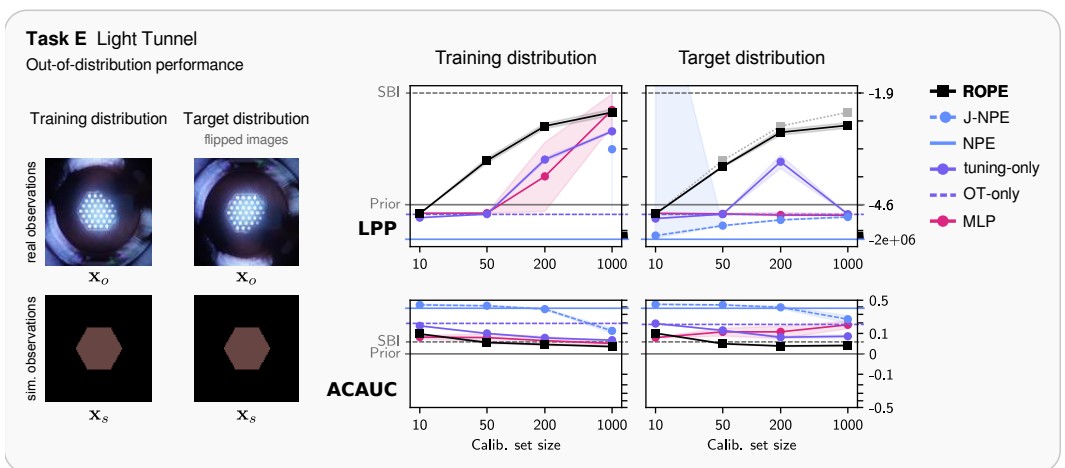

Figure 6: Out-of-distribution performance of RoPE and some baselines. We train RoPE and other baselines on the same light-tunnel data as in task E (training distribution), but apply it to test sets originating from a target distribution where the real-world images are flipped vertically. We compare the performance on test sets from both distributions, showing the LPP and ACAUC scores for each method. For comparison, in the right plot we show again the LPP curve (light gray, dotted) attained by RoPE under the training distribution. The performance of RoPE is barely affected as it cannot exploit any signal in the real images ($\mathbf{x}_o$) beyond what is encoded in the simulator, and the simulator output ($\mathbf{x}_s$) is invariant to the transformation we consider. Because NPE is not trained on real observations, its performance, although poor, also remains virtually unchanged. On the other hand, the performance of MLP and J-NPE drops in the target distribution, as these methods are not limited in what information they can exploit from the real observations on which they are trained, potentially learning shortcuts that are not present in the target distribution. This results demonstrate that if the simulator embeds the right invariances, our modeling assumption $\mathbf{x}_o \perp \theta \mid \mathbf{x}_s$ can be favorable to out-of-distribution generalization.

# E    OPTIMAL TRANSPORT COUPLING AS A JOINT DISTRIBUTION

With our conditional independence assumption, the problem of modeling $p(\mathbf{x}_o \mid \theta)$ reduces to modeling $p(\mathbf{x}_o \mid \mathbf{x}_s)$ instead. If we assume the prior well-specified, this task is equivalent to modeling $p(\mathbf{x}_o, \mathbf{x}_s)$ under the constraint that the corresponding marginal $p(\mathbf{x}_s) = \int p(\mathbf{x}_s, \mathbf{x}_o) d\mathbf{x}_o$ equals $\int p(\theta) p(\mathbf{x}_s \mid \theta) d\theta$. By construction, the OT coupling, $\pi^\star$, respects the constraint on the marginals, $\int \pi^\star(\mathbf{x}_s, \mathbf{x}_o) d\mathbf{x}_o = p(\mathbf{x}_s)$ and $\int \pi^\star(\mathbf{x}_s, \mathbf{x}_o) d\mathbf{x}_s = p(\mathbf{x}_o)$, and the exact instantiation $\pi^\star$ depends also on the chosen cost function which can always be defined to yield any given conditional $p(\mathbf{x}_o \mid \mathbf{x}_s)$ that respects the constraint $\int p(\mathbf{x}_o \mid \mathbf{x}_s) p(\mathbf{x}_s) d\mathbf{x}_s = p(\mathbf{x}_o)$. $\pi^*$ can thus model the "right" posterior, provided the right cost function is used. In the case, where the prior cannot be trusted, we suggest to use $\tau < 1$ and relax the OT formulation. In this case, we only enforce that all elements of $p(x_o)$ are matched to a subset of the elements of $p(x_s)$. This implicitly assumes that the assumed prior $p(\theta)$ is overly conservative and covers $p^\star(\theta)$. We believe this is a reasonable assumption as it is often easy to derive physical bounds for the parameter values and use a uniform distribution.

## F SELF-CALIBRATION PROPERTY

We say RoPE is self-calibrating because, by design, the posterior distribution marginalized over observations tends to the prior as the number of simulation increases, that is,

$$\int_{\mathcal{X}} \tilde{p}(\theta \mid \mathbf{x}_o)p(\mathbf{x}_o)d\mathbf{x}_o = p(\theta). \tag{7}$$

This property is also called marginal calibration, and is a necessary condition for a posterior estimation method to be calibrated. Considering NPE, $\tilde{p}(\theta \mid \mathbf{x}_s)$, is marginally calibrated and observations $\mathbf{x}_o$ are generated from the assumed prior, that is sampled from an unknown distribution $p(\mathbf{x}_o) = \int p(\mathbf{x}_o \mid \theta)p(\theta)$, we can show RoPE is marginally calibrated. Indeed, considering the Monte-Carlo approximation of the marginalized posterior distribution over the test set $\mathcal{D}_o := \{\mathbf{x}_o^i\}_{i=1}^{N_o}$, we have,

$$\int_{\mathcal{X}} \tilde{p}(\theta \mid \mathbf{x}_o)p(\mathbf{x}_o)d\mathbf{x}_o = \mathbb{E}_{p(\mathbf{x}_o)}[\tilde{p}(\theta \mid \mathbf{x}_o)] \tag{8}$$

$$\approx \frac{1}{N_o} \sum_{i=1}^{N_o} \tilde{p}(\theta \mid \mathbf{x}_o^i) \tag{9}$$

$$= \frac{1}{N_o} \sum_{i=1}^{N_o} \sum_{j=1}^{N_s} N_o P_{ij}^{\star} \tilde{p}(\theta \mid \mathbf{x}_s^j) \tag{10}$$

$$= \sum_{j=1}^{N_s} \left[ \sum_{i=1}^{N_o} P_{ij}^{\star} \right] \tilde{p}(\theta \mid \mathbf{x}_s^j) \tag{11}$$

$$= \frac{1}{N_s} \sum_{j=1}^{N_s} \tilde{p}(\theta \mid \mathbf{x}_s^j) \tag{12}$$

$$\approx p(\theta), \tag{13}$$

where we use the definition of the transport matrix to get $\sum_{i=1}^{N_o} P_{ij}^{\star} = \frac{1}{N_s}$. The last approximation tends to be exact as the number of simulations increases, if the NPE is marginally calibrated.

## G LEARNING MINIMAL SUFFICIENT STATISTICS WITH NEURAL POSTERIOR ESTIMATION

We now discuss why NPE may learn a minimal sufficient statistic under perfect training. First, under a sufficiently large validation set, NPE's objective function is only optimal on the validation set if NPE models the true posterior as defined implicitly by the prior $p(\theta)$ and the likelihood corresponding to the simulator $S$. This consistency has been proven in (Papamakarios & Murray, 2016) and is the motivation to use such an objective when estimating density. Second, some normalizing flows, such as autoregressive UMNN flows (Wehenkel & Louppe, 2019), are universal approximators of continuous densities. In addition, neural networks are also universal function approximators. As such, we can claim that it is always possible to parameterize the NCDE $p_\theta(\theta \mid \mathbf{h}_\omega(\mathbf{x}))$ such that the class of functions its parameters represent contains the true posterior. We directly observe that $\mathbf{x}$ is only used by the NCDE through $\mathbf{h}_\omega(\mathbf{x})$. Thus, under perfect training $p_{\theta^\star}(\theta \mid \mathbf{h}_{\omega^\star}(\mathbf{x})) = p(\theta \mid \mathbf{x})$ and $\mathbf{h}_{\omega^\star}(\mathbf{x})$ is a sufficient statistic for $\theta$ given $\mathbf{x}$ under the simulator's model.

Without additional constraints, we cannot claim anything about the minimality of $\mathbf{h}_{\omega^\star}(\mathbf{x})$. Nevertheless, we can enforce the neural network $\mathbf{h}_{\omega^\star}(\mathbf{x})$ to have an information bottleneck and thus reduce the information carried. In practice, we choose the output dimension of $\mathbf{h}_{\omega^\star}(\mathbf{x})$ so that the NCDE achieves optimal performance on the test set. Because in the context of SBI we can generate as many (simulated) samples as needed, we can obtain estimators that closely approach the simulation's posterior and a minimal sufficient statistic.

## H COMPUTATIONAL COST OF RoPE

Running NPE is broadly recognized as having a low computational cost: once the upfront training is complete, the cost of inverting the normalizing flow to sample from the posterior during inference becomes negligible as the number of test observations increases. This makes NPE more efficient than methods like Approximate Bayesian Computation or Markov Chain Monte Carlo (when the simulator allows likelihood evaluation). RoPE introduces additional computational costs on top of running NPE: (1) the OT coupling computation, i.e., solving Equation 2, and (2) obtaining samples from the estimated posterior distributions, to compute the posterior estimate defined in Equation 5. The computational cost of solving the transport problem with the Sinkhorn algorithm (Cuturi, 2013) is quadratic in the number of real-world observations. The sampling step has a negligible cost as it directly sub-samples from the set of points generated with NPE.

In our experiments, solving the OT optimization for 2000 test examples takes less than a minute on an M1 MacBook Pro. Sampling from the mixture of posterior distributions involves caching 10,000 samples for each simulation and generating 5,000 samples by sub-sampling from the mixture using the OT coupling matrix. This caching process takes under three minutes, and is comparable to the cost of running NPE alone.

Extending RoPE to handle larger test sets or an online setting (processing test examples one at a time) is outside the scope of this work. Nevertheless, mehtods like Neural OT (e.g., (Makkuva et al., 2020)) and online Sinkhorn (Mensch & Peyré, 2020) should provide good solutions to make RoPE fully amortized.

## I EXPERIMENTAL SETUP

In this section, we provide more details on our experiments. For completeness, we provide details on the neural architectures and training hyperparameters. However, we encourage the reader interested in reproducing our experiments to examine our code directly (a link to the code will be made available in the public version of the paper).

For all methods training on calibration set we keep always keep $20\%$ of the calibration to monitor validation performance and we select the best model based on this metric.

For the MLP we use the same architecture as the NSE for all our experiments and optimize its parameters on the calibration set with Adam and a learning rate equal to 0.0003, we select the best model based on the LPP attributed to the validation subset of the calibration set.

**Computing the SBI baseline.** We take the ground-truth labels $\{(\theta^i)\}_{i=1}^N$ from the test set $\{(\theta^i, \mathbf{x}_o^i)\}_{i=1}^N$ on which we compute all the metrics for Figure 2; for each label $\theta^i$, we simulate a synthetic observation $\mathbf{x}_s^i := \mathcal{S}(\theta^i)$, collecting them into a "synthetic" test set $\{(\theta^i, \mathbf{x}_s^i)\}_{i=1}^N$; then, we apply to it the NSE+NPE pipeline (simulated posterior in Figure 1, right) to obtain the posterior estimates which we then evaluate. In this way, the baseline represents the performance we would hope to achieve if there was no misspecification and the simulator perfectly replicated the real observations (up to the stochasticity of the simulator itself).

### I.1 TASK A: CS & TASK B: SIR

**Task A (synthetic): CS.** We reproduce the cancer and stromal cell development benchmark from Ward et al. (2022). The simulator emulates the development of cancer and stromal cells in a 2D environment as a function of three Poisson rate parameters $(\lambda_c, \lambda_p, \lambda_d)$. The observations are vectors composed of the number of cancer and stromal cells and the mean and maximum distance between stromal cells and their nearest cancer cell. Synthetic misspecification is introduced by removing cancer cells that are too close to their generating parent.

**Task B (synthetic): SIR.** We also use the stochastic epidemic model from Ward et al. (2022), which describes epidemic dynamics through the infection rate $\beta$ and recovery rate $\gamma$. Each observation is a vector composed of the mean, median, and maximum number of infections, the day of occurrence of the maximum number of infections, the day at which half the total number of infections was reached, and the mean auto-correlation (lag 1) of the infections. Misspecification is a delay in weekend infection counts, of which $5\%$ are added to the count of the following Monday.

We refer the reader to Ward et al. (2022) for more details about the simulator and prior distribution. We use the exact same setting as theirs.

### NEURAL ARCHITECTURE & TRAINING HYPERPARAMETERS

For all methods we use the same backbone MLP as the NSE with ReLU activations and layers composed of $[4K, 16K, 16K, 12K, 3K]$ neurons, where $K$ is the dimensionality of $\theta$. The NF is a 1-step UMNN-MAF (Wehenkel & Louppe, 2019) with $[100, 100, 100]$ neurons for both the autoregressive conditioner and normalizer. For NNPE, we train the UMNN-MAF on simulations poluted by Spike and Slab errors. We train models with Adam and a learning rate equal to $0.0005$ and all other parameters set to default. We optimize the SBI model for $10^6$ gradient steps and select the best model on random validation sets containing $10^5$ simulations.

### I.2 TASK C: PENDULUM

#### DESCRIPTION

The first task is inspired from the damped pendulum benchmark commonly used to assess hybrid learning algorithms. Given a 2D physical parameter $\theta := [\omega_0, A]$, where $\omega_0 \in \mathbb{R}^+$ denotes the fundamental frequency and $A \in \mathbb{R}^+$ the amplitude of a friction-less pendulum, the simulator generates the horizontal position of the pendulum at 200 discrete times during uniformly sampled in a 10 seconds interval as

$$\mathbf{x}_s := [\theta(t = 0), \ldots, \theta(t = 10s)] \in \mathbb{R}^{200}$$
$$\text{where } \theta(t) = A \cos(\omega_0 t + \varphi) \quad \varphi \sim \mathbb{U}(-\pi, \pi). \tag{14}$$

The relationship between the parameters and the simulation is thus stochastic as $\varphi$ accounts for an unknown phase shift when the measurements start. We generate real-world observations synthetically by replacing $\theta(t)$ from (14) by

$$\tilde{\theta}(t) = e^{\alpha t} A \cos(\omega_0 t + \varphi) \quad \varphi \sim \mathbb{U}[-\pi, \pi] \quad \alpha \sim \mathbb{U}[0, 1],$$

where $\alpha$ represents the effect of friction. We also add Gaussian noise on both simulated and real-world data to represent the inaccuracy of a sensor measuring the pendulum's position. The prior distribution is a product of uniform distribution, $p(\theta := [\omega_0, A]) = \mathcal{U}[0, 3] \times \mathcal{U}[0.5, 10]$.

### NEURAL ARCHITECTURE & TRAINING HYPERPARAMETERS

**Neural Posterior Estimator.** The NSE is a 1D convolutional neural network, with the architecture described in Algorithm 2. The NCDE is a one-step discrete normalizing flow with an autoregressive

---

**Algorithm 2** Convolutional Neural Network for Tasks A and D.

---

1: $\text{Conv1d}(1, 16, 3, 1, \text{dilation} = 2, \text{padding} = 1)$
2: $\text{ReLU}()$
3: $\text{Conv1d}(16, 64, 3, 2, \text{dilation} = 2, \text{padding} = 1)$
4: $\text{ReLU}()$
5: $\text{AvgPool1d}(3, 1)$
6: $\text{Conv1d}(64, 128, 3, 1, \text{dilation} = 2, \text{padding} = 1)$
7: $\text{ReLU}()$
8: $\text{Conv1d}(128, 128, 3, 2, \text{dilation} = 2, \text{padding} = 1)$
9: $\text{ReLU}()$
10: $\text{AvgPool1d}(3, 1)$
11: $\text{Conv1d}(128, 128, 3, 1, \text{dilation} = 2, \text{padding} = 1)$
12: $\text{ReLU}()$
13: $\text{Conv1d}(128, 128, 3, 2, \text{dilation} = 2, \text{padding} = 1)$
14: $\text{ReLU}()$
15: $\text{AvgPool1d}(3, 1)$
16: $\text{Conv1d}(128, 128, 3, 1, \text{dilation} = 2, \text{padding} = 1)$
17: $\text{ReLU}()$
18: $\text{Flatten}()$
19: $\text{Linear}(2048, 512)$
20: $\text{ReLU}()$
21: $\text{Linear}(512, 128)$
22: $\text{ReLU}()$
23: $\text{Linear}(128, 32)$
24: $\text{ReLU}()$
25: $\text{Linear}(32, 10)$

---

**Algorithm 3** UNet1D Architecture

1: Unet1D :
2:   Encoder1D :
3:     Block(in_channels $= 1$, out_channels $= 64$)
4:     Block(in_channels $= 64$, out_channels $= 128$)
5:     Block(in_channels $= 128$, out_channels $= 256$)
6:     Block(in_channels $= 256$, out_channels $= 512$)
7:     Block(in_channels $= 512$, out_channels $= 1024$)
8:     MaxPool1d(2)
9:   Decoder1D :
10:     ConvTranspose1d(1024 $+ 5$, $512$, $2$, stride $= 2$)
11:     Block(in_channels $= 1024$, out_channels $= 512$)
12:     ConvTranspose1d($512, 256, 2$, stride $= 2$)
13:     Block(in_channels $= 512$, out_channels $= 256$)
14:     ConvTranspose1d($256, 128, 2$, stride $= 2$)
15:     Block(in_channels $= 256$, out_channels $= 128$)
16:     ConvTranspose1d($128, 64, 2$, stride $= 2$)
17:     Block(in_channels $= 128$, out_channels $= 64$)
18:     ConvTranspose1d($64, 1, 2$, stride $= 2$)
19:     Block(in_channels $= 64$, out_channels $= 1$)
20:     Conv1d($64, 1, 1$)

**Algorithm 4** Block1D(in_channels, out_channels)

1: Conv1d(in_channels, out_channels, kernel_size=3, padding=1)
2: ReLU()
3: Conv1d(out_channels, out_channels, kernel_size=3, padding=1)
4: ReLU()

**Algorithm 5** 2D Convolutional Neural Network

1: Conv2d(3, 64, 3, 2, dilation=1), ReLU()
2: Conv2d(64, 128, 3, 2, dilation=1), ReLU()
3: MaxPool2d(3)
4: Conv2d(128, 128, 3, 2, dilation=1), ReLU()
5: Conv2d(128, 64, 1, 1, dilation=1), ReLU()
6: Conv2d(64, 3, 1, 1, dilation=1), ReLU()
7: Flatten()
8: Linear(27, 100), ReLU()
9: Linear(100, 20)

conditioner and a UMNN (Wehenkel & Louppe, 2019) as the normalizer. The autoregressive conditioner is a MADE with ReLU activation and 3 layers of 100 neurons that output a 10 dimensional vector to the UMNN. The UMNN has an integrand net with 3 layers of 100 neurons with ReLU activations. For training the NPE, we use a batch size of 100 and a learning factor equal to 1e-4. NPE is trained until convergence. Other parameters are set to default values and should marginally impact the NPE obtained.

**RoPE NSE.** We have selected the best NPE based on the validation set with 10000 examples generated with the simulator. The NPE is fixed to one best-of-all model. We fine-tune the NCDE with a learning rate equal to 1e-5 for 5000 gradient steps on 80% the full calibration set. We use a 1-sample Monte Carlo estimate of the expectation in (6).

**J-NPE.** To train J-NPE, we simply randomly use a batch composed of 50% of simulated pairs $(\theta, \mathbf{x}_s)$ and of 50% $(\theta, \mathbf{x}_o)$ from the calibration set. We use the same architecture and hyper-parameters as the SBI NPE. The best model is selected based on the best training set performance. We do 50 epochs with 50000 simulated examples for each epoch. The batch size is 100.

**HVAE.** For the HVAE, we re-use the NPE model as the physics encoder and replace the decoder with a deterministic version of the simulator, thus removing the Gaussian noise on a random phase shift. In addition, we follow the approach of Takeishi & Kalousis (2021) and have 1) a real-world encoder that maps $\mathbf{x}_o$ to $\mathbf{z}_a$, 2) a reality-to-physics encoder, and 3) a physics-to-reality decoder. The real-world encoder has the same architecture as the NSE of the NPE and outputs the mean and log-variance of a 5D latent vector $\mathbf{z}_a$. The reality-to-physics and physics-to-reality also have the same architectures and are two conditional 1D U-Net with neural network architecture described in Algorithm 3.

To train the HVAE, we freeze the parameters of the NPE and optimizes the ELBO as well as a calibration loss that evaluates the likelihood assigned to the true physical parameters. All distributions are parameterized by Gaussian with mean and log-variance predicted by the neural networks. We do not use any additional losses as we expect constraining NPE and using the calibration set should already provide the necessary support to use the physics in a meaningful way. The HVAE is trained on the 2000 test examples as it is the only real-world data, calibration set aside, that we have access to. We use a batch size equal to 100 and a learning rate equal to 1e-3. We believe obtaining a better HVAE is possible. However, we emphasize the complexity of setting up a good HVAE for the only purpose of statistical inference over parameters.

### DATASETS

For this task, we can generate samples $(\theta, \mathbf{x}_s)$ on the fly to train the NPE. The calibration and test sets are also generated randomly by sampling from the prior distribution and using the damped pendulum simulator.

### I.3 TASK D: HEMODYNAMICS

### DESCRIPTION

Inspired by Wehenkel et al. (2023), we define the task of inferring important cardiovascular parameters from normalized arterial pressure waveforms measured at the radial artery. The simulator uses many physiological parameters that modulates the heart function, physical properties of the 116 main arterial segments, and behavior of the vascular beds. Our inference concerns two parameters of the heart function, $\theta := [\mathrm{SV}, \mathrm{LVET}]$, the stroke volume (SV) is the amount pumped out from the left ventricle over the heart beat modeled, and the left ventricular ejection time (LVET) is the time interval between opening and closure of the aortic valve. Other parameters, such as the heart rate or arteries' stiffness, are considered as nuisance effects and are randomly sampled from a realistic population distribution. An additional source of randomness is added by modeling measurement errors with a white Gaussian noise and randomizing the starting recording time with respect to the cardiac cycle. The simulator produces 8-second timeseries $\mathbf{x}_t \in \mathbb{R}^{1000}$ sampled at $125Hz$. As synthetic misspecification, the simulator assumes all arteries have the same length over the population considered, whereas "real-world" data are artificially generated by also varying the length of arteries and account for the effect of human's height. The simulator is based on the openBF PDE solver (Melis, 2017) specialized for hemodynamics, which is not differentiable and takes approximately one minute to simulate one sample on a standard CPU. This synthetic tasks represent a common scenario in which a simulator, although faithful to the effect of certain parameters, misses additional degrees of freedom that exists for the real-world data.

### NEURAL ARCHITECTURE & TRAINING HYPERPARAMETERS

---

**Algorithm 6** CNN Architecture for Task C.

---

 1: Conv1d(1, 16, 3, 1, dilation=2, padding=1), ReLU()
 2: Conv1d(16, 64, 3, 2, dilation=2, padding=1), ReLU()
 3: AvgPool1d(4, 2)
 4: Conv1d(64, 128, 3, 1, dilation=2, padding=1), ReLU()
 5: Conv1d(128, 128, 3, 2, dilation=2, padding=1), ReLU()
 6: AvgPool1d(4, 2)
 7: Conv1d(128, 128, 3, 1, dilation=2, padding=1), ReLU()
 8: Conv1d(128, 128, 3, 2, dilation=2, padding=1), ReLU()
 9: AvgPool1d(4, 1)
10: Conv1d(128, 128, 3, 1, dilation=2, padding=1), ReLU()
11: Flatten()
12: Linear(1024, 512), ReLU()
13: Linear(512, 128), ReLU()
14: Linear(128, 32), ReLU()
15: Linear(32, 5)

---

**Neural Posterior Estimator.** The NSE is the 1D convolutional neural network described in Algorithm 6. The NCDE is a 5-step discrete normalizing flow with an autoregressive conditioner and affine normalizers. Each of the 5 autoregressive conditioners is a MADE with ReLU activations and 4 layers of 300 neurons that output 4 dimensional vectors used to parameterize the affine transformations. For training the NPE, we use a batch size of 100 and a learning factor equal to 5e-4. NPE is trained until convergence. Other parameters are set to default values and should marginally impact the NPE obtained.

**RoPE NSE.** We have selected the best NPE based on the validation set with 2000 examples generated with the simulator. The NPE is fixed to one best-of-all model. We fine-tune the NCDE with a learning rate equal to 1e-5 for 2000 gradient steps on 80% of calibration set. We use a 1-sample Monte Carlo estimate of the expectation in (6).

**J-NPE.** To train J-NPE, we simply randomly use a batch composed of 50% of simulated pairs $(\theta, \mathbf{x}_s)$ and of 50% $(\theta, \mathbf{x}_o)$ from the calibration set. We use the same architecture and hyper-parameters as the SBI NPE. The best model is selected based on the best training set performance. We do 50 epochs with 6000 simulated examples for each epoch. The batch size is 100.

**HVAE.** There is no HVAE for this experiment as the simulator is non-differentiable.

DATASETS

For this task, we cannot generate samples $(\theta, \mathbf{x}_s)$ on the fly to train the NPE. For the purpose of this experiment, we have generated 10000 simulations and real-world observations. Our fine-tuning strategy approximates (6) by finding the simulations with the closest parameter value.

I.4    TASK E: LIGHT TUNNEL

DESCRIPTION

We use one of the light-tunnel datasets from the causal chamber project (Gamella et al., 2024, causalchamber.org). In particular, we use the data from the `ap_1.8_iso_500.0_ss_0.005` experiment in the `lt_camera_v1` dataset. The light tunnel is an elongated chamber with a controllable light source at one end, two linear polarizers mounted on rotating frames, and a camera that takes images of the light source through the polarizers. We refer the reader to Gamella et al. (2024, Figure 2) for a complete schematic. Our task consists of predicting the color setting of the light source $((R, G, B) \in [0, 255]^3)$ and the dimming effect of the linear polarizers $\alpha \in [0, 1]$ from the captured images. As a misspecified simulator of the image-generating process, we adopt the simple model described in Gamella et al. (2024, Model F1, Appendix D). A Python implementation is available through the `causalchamber` package (`models.model_f1`); visit causalchamber.org for more details. As input, the simulator takes the parameters $\theta := [R, G, B, \alpha]$ and produces an image consisting of a hexagon roughly the size of the light source, with an RGB color vector equal to $[\alpha R, \alpha G, \alpha B]$. The factor $\alpha := \cos^2(\theta_1 - \theta_2)$, where $\theta_1, \theta_2$ denote the angles of the two polarizers, corresponds to Malus' law (e.g. , Collett, 2005), which models the dimming effect of the polarizers as a function of their relative angle. Besides the obvious misspecification with respect to image realism (see Figure 2), the model ignores other important physical aspects, such as the spectral response of the camera sensor or the non-uniform effect of the polarizers on the different colors—more details can be found in Gamella et al. (2024, Appendix D.IV.2.2). The prior is uniform over colors and polarizer angles, which leads to a non-uniform prior over the dimming effect $\alpha$.

NEURAL ARCHITECTURE & TRAINING HYPERPARAMETERS

**Neural Posterior Estimator.** The NSE is the 2D convolutional neural network described by Algorithm 5.

The NCDE is also a one-step discrete normalizing flow with an autoregressive conditioner and a UMNN (Wehenkel & Louppe, 2019) as the normalizer. The autoregressive conditioner is a MADE with ReLU activation and 3 layers of 500 neurons that outputs a 10 dimensional vector to the UMNN. The UMNN has an integrand net with 4 layers of 150 neurons with ReLU activations. For training the NPE, we use a batch size of 100 and a learning factor equal to 5e-4. NPE is trained until convergence. Other parameters are set to default values and should marginally impact the NPE obtained.

**Algorithm 7** 2D UNet

1: Encoder2D:
2:    Block2D(in_channels=3, out_channels=64)
3:    Block2D(in_channels=64, out_channels=128)

4:    Block2D(in_channels=128, out_channels=256)
5:    Block2D(in_channels=256, out_channels=512)
6:    Block2D(in_channels=512, out_channels=1024)
7:    MaxPool2d(2)
8: Decoder2D:
9:    ConvTranspose2d(1024 + 5, 512, 2, stride=2)
10:    Block2D(in_channels=1024, out_channels=512)
11:    ConvTranspose2d(512, 256, 2, stride=2)
12:    Block2D(in_channels=512, out_channels=256)
13:    ConvTranspose2d(256, 128, 2, stride=2)
14:    Block2D(in_channels=256, out_channels=128)
15:    ConvTranspose2d(128, 64, 2, stride=2)
16:    Block2D(in_channels=128, out_channels=64)

17:    ConvTranspose2d(64, 1, 2, stride=2)
18:    Block2D(in_channels=64, out_channels=1)
19:    Conv2d(64, 1, 1)

**Algorithm 8** Block2D(in_channels, out_channels)

1: Conv2d(in_channels, out_channels, kernel_size=3, padding=1, bias=False)
2: BatchNorm2d(num_features=out_channels)

3: ReLU(inplace=True)
4: Conv2d(out_channels, out_channels, kernel_size=3, padding=1, bias=False)
5: BatchNorm2d(num_features=out_channels)

6: ReLU(inplace=True)

**RoPE NSE.** We have selected the best NPE based on the validation set with 10000 examples generated with the simulator. The NPE is fixed to one best-of-all model. We fine-tune the NCDE with a learning rate equal to 1e-4 for 2000 gradient steps on on 80% of the calibration set. We use a 1-sample Monte Carlo estimate of the expectation in (6).

**J-NPE.** To train J-NPE, we simply randomly use a batch composed of 50% of simulated pairs $(\theta, \mathbf{x}_s)$ and of 50% $(\theta, \mathbf{x}_o)$ from the calibration set. We use the same architecture and hyper-parameters as the SBI NPE. The best model is selected based on the best training set performance. We do 50 epochs with 1000 simulated examples for each epoch. Simulations are generated randomly for each batch by sampling the prior and simulating for the corresponding parameters. The batch size is 100.

**HVAE.** For the HVAE, we re-use the NPE model as the physics encoder and use the simulator as is as it is differentiable without additional effort. In addition, we follow the approach of Takeishi & Kalousis (2021) and have 1) a real-world encoder that maps $\mathbf{x}_o$ to $\mathbf{z}_a$, 2) a reality-to-physics encoder, and 3) a physics-to-reality decoder. The real-world encoder has the same architecture as the NSE of the NPE and outputs the mean and log-variance of a 5D latent vector $\mathbf{z}_a$. The reality-to-physics and physics-to-reality also have the same architectures and are two conditional 2D U-Net with the architecture described by Algorithm 7.

To train the HVAE, we freeze the parameters of the NPE and optimizes the ELBO as well as a calibration loss that evaluates the likelihood assigned to the true physical parameters. All distributions are parameterized by Gaussian with mean and log-variance predicted by the neural networks. We do not use any additional losses as we expect constraining NPE and using the calibration set should already provide the necessary support to use the physics in a meaningful way. The HVAE is trained on the 2000 test examples as it is the only real-world data, calibration set aside, that we have access to. We use a batch size equal to 100 and a learning rate equal to 1e-3. We believe obtaining a better HVAE is possible. However, we emphasize the complexity of setting up a good HVAE for the only purpose of statistical inference over parameters.

DATASETS

For this task, we can generate samples $(\theta, \mathbf{x}_s)$ on the fly to train the NPE. However, the calibration and test sets are real-world data. We ensure there is not overlap between calibration and test set. The is no randomization and the test set is constant for all experiments, the calibration set are also fixed for a given calibration set size.

## I.5 TASK F: WIND TUNNEL

DESCRIPTION

We use one of the wind-tunnel datasets from the causal chamber project (Gamella et al., 2024, `causalchamber.org`). In particular, we use the data from the `load_out_0.5_osr_downwind_4` experiment in the `wt_intake_impulse_v1` dataset. The tunnel is a chamber with two controllable fans that push air through it and barometers that measure air pressure at different locations. A hatch precisely controls the area of an additional opening to the outside (see Gamella et al., 2024, Figure 2). The data is a collection of pressure curves that result from applying a short impulse to the intake fan load and measuring the change in air pressure using one of the barometers inside the tunnel. Our inference task consists of predicting the hatch position, $\theta := [H] \in [0, 45]$ given a pressure curve (see Figure 2). As a simulator model, we combine the models A2 and C3 described in Gamella et al. (2024, Appendix D); we numerically solve the ODE in model A2, and add stochastic components to simulate the sensor noise and the unknown time point at which the impulse is applied. This results in the simulator being neither differentiable nor deterministic. A Python implementation of the complete simulator is available in the `causalchamber` package (`models.simulator_a2_c3`); visit `causalchamber.org` for more details. Misspecification arises from the many simplifying assumptions needed to model the complex dynamics of the airflow inside the tunnel—more details can be found in Gamella et al. (2024, Appendix D.IV.1.2).

**Neural Posterior Estimator.** The NSE and NCDE have the same 1D convolutional neural network as for Task A. For training the NPE, we use a batch size of 100 and a learning factor equal to 5e-4. NPE is trained until convergence. Other parameters are set to default values and should marginally impact the NPE obtained.

**RoPE NSE.** We have selected the best NPE based on the validation set with 10000 examples generated with the simulator. The NPE is fixed to one best-of-all model. We fine-tune the NCDE with a learning rate equal to 1e-4 for 20000 gradient steps on on 80% of the calibration set. We use a 1-sample Monte Carlo estimate of the expectation in (6).

**J-NPE.** To train J-NPE, we simply randomly use a batch composed of 50% of simulated pairs $(\theta, \mathbf{x}_s)$ and of 50% $(\theta, \mathbf{x}_o)$ from the calibration set. We use the same architecture and hyper-parameters as the SBI NPE. The best model is selected based on the best training set performance. We do 50 epochs with 10000 simulated examples for each epoch. The batch size is 100.

**HVAE.** There is no HVAE for this experiment as the simulator is non-differentiable.

DATASETS

For this task, although slightly slower than Task A and B, we can generate samples $(\theta, \mathbf{x}_s)$ on the fly to train the NPE. However, the calibration and test sets are real-world data. We ensure no overlap between the two sets for all calibration set sizes. All sets are fixed for all experiments.

## J  COMPUTING ACAUC

---

**Algorithm 9** Statistical Calibration of Posterior Distribution

---

**Input:** Dataset of pairs $\mathcal{D} = \{(\theta^i, \mathbf{x}^i)\}$, Posterior estimator $\tilde{p}(\theta \mid \mathbf{x})$, Number of samples $N$.
**Output:** ACAUC

1:  AVG_CALIBRATION $= 0$
2:  **for** $k \in \{1, \ldots, K\})$ **do**
3:      Initialize an empty list CredLevels
4:      **for** $(\theta^i, \mathbf{x}^i) \in \mathcal{D}$ **do**
5:          Initialize an empty list Samples
6:          **for** $j = 1$ to $M$ **do**
7:              Sample $\theta^j$ from $\tilde{p}(\theta \mid \mathbf{x}^i)$
8:              Append $\theta^j$ to Samples
9:          **end for**
10:         Sort Samples
11:         Compute the rank (position in ascending order) $r$ of $\theta$ in Samples
12:         Set CredLevels $= \frac{r}{N}$
13:         Append CredLevel to CredLevels
14:     **end for**
15:     Sort CredLevels
16:     CALIBRATION $= \sum_{i=1}^{N}$ CredLevels$[i] - \frac{i}{N}$
17:     AVG_CALIBRATION $=$ AVG_CALIBRATION $+ \frac{\text{CALIBRATION}}{K}$
18: **end for**
**Return:** AVG_CALIBRATION

---

## K  ADDITIONAL RESULTS

### K.1  CORNER AND CALIBRATION PLOTS

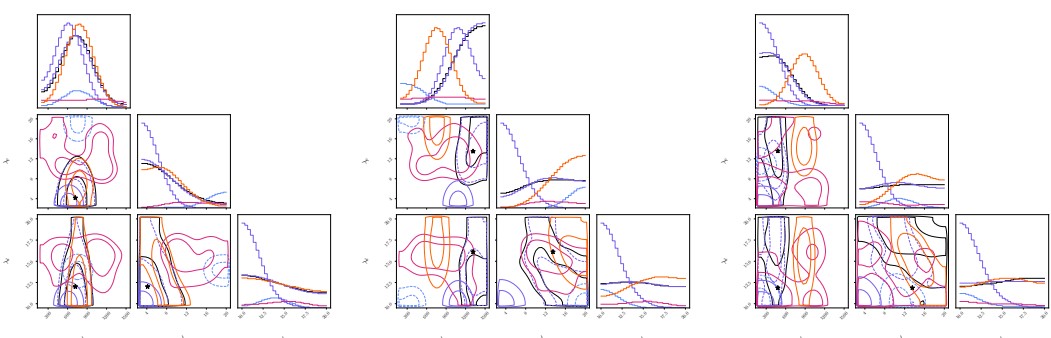

Figure 7: Three corner plots for task A with a calibration set with 50 samples.

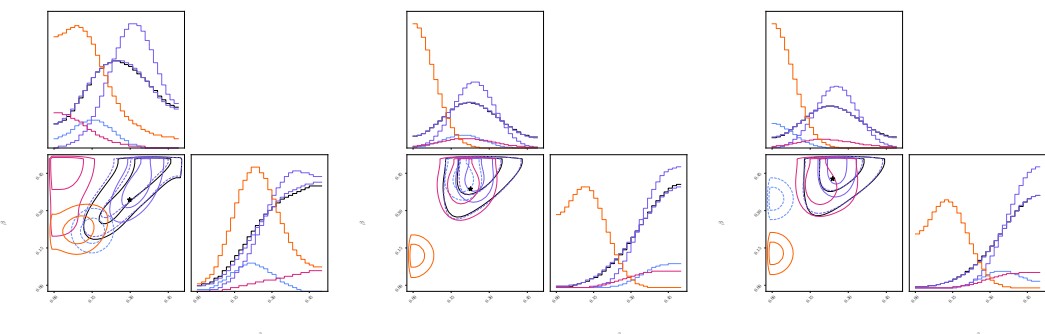

Figure 8: Three corner plots for task B with a calibration set with 50 samples.

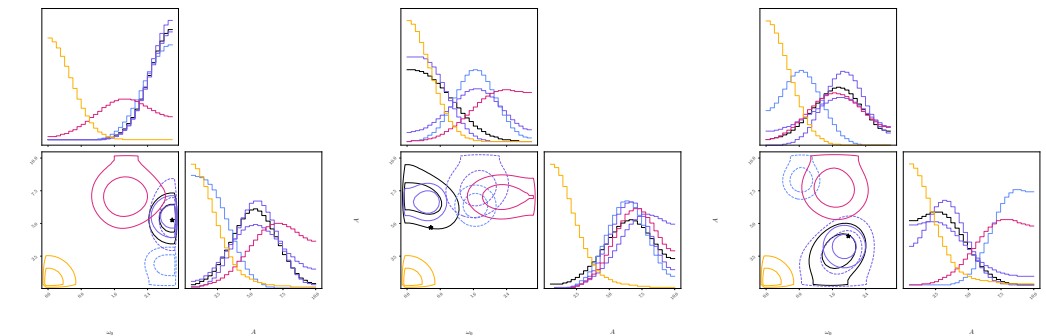

Figure 9: Three corner plots for task C with a calibration set with 50 samples.

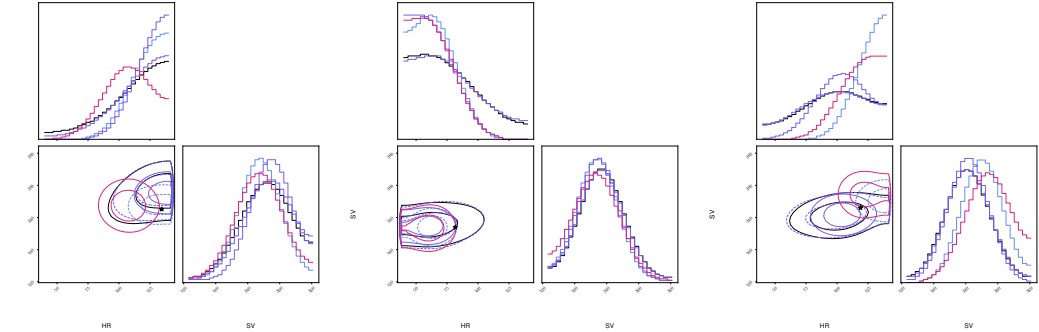

Figure 10: Three corner plots for task D with a calibration set with 50 samples.

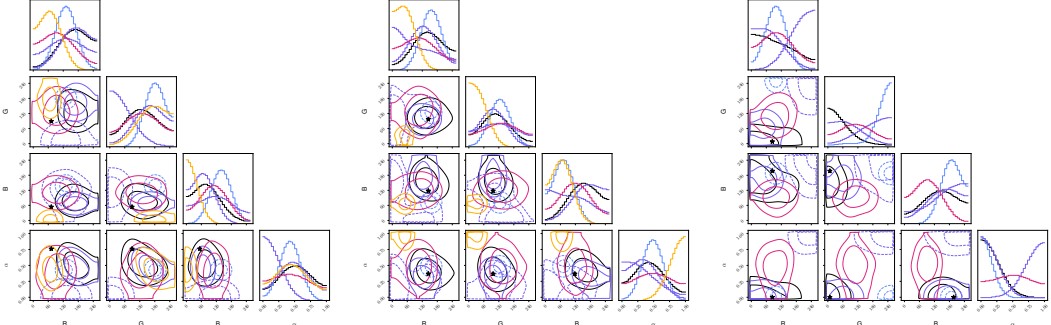

Figure 11: Three corner plots for task E with a calibration set with 50 samples.

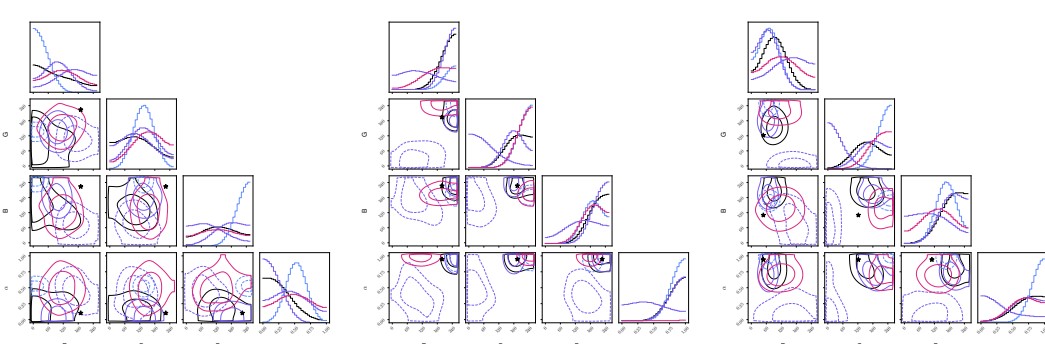

Figure 12: Three corner plots for task E with distribution shift with a calibration set with 50 samples.

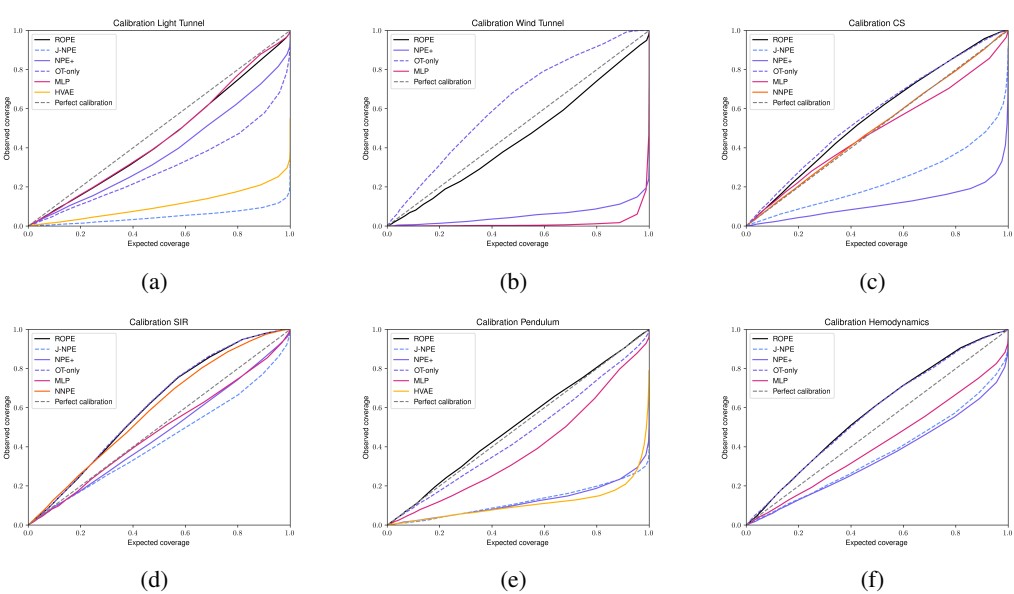

Figure 13: Calibration plots of the different methods on the 6 benchmarks, the coverage at each level is the average of the coverage of the marginal distributions. Each color indicates a different algorithm and the opacity is proportional to the size of the calibration set which ranges from 10 to 1000. We observe that RoPE and OT-only are consistently well calibrated for.