# OpenReview forum: "Addressing Misspecification in Simulation-based Inference through Data-driven Calibration"
_ICLR.cc/2025/Conference — Submitted to ICLR 2025_

### Official Review · Reviewer_MKFY · 2024-10-21

**Soundness:** 3
**Presentation:** 2
**Contribution:** 3
**Rating:** 6
**Confidence:** 3

**Summary:**

The paper details with the important issue of making neural posterior estimation methods more robust against model misspecification. The authors suggest to use a small set of labeled real-world data to calibrate the posterior inference in the face of a misspecified simulator. At the moment, I see some important weakness that make me cautios at recommending acceptance at this stage. But I am open and happy to change my mind during the discussion period.

After the discussion period, I have updated my score from 5 to 6.

**Strengths:**

- The paper investigates an important issue.
- The goals of this paper are clearly explained
- Detailed examples and ablation studies are performed.

**Weaknesses:**

- The main weakness I see is the following: How do we gain access to the *labeled* real-world observations? This seems like quite a strong assumption. I would need to understand if and when this assumption is justified before I would consider recommending acceptance.
- As an expert in SBI, I struggle to understand the introduction of optimal transport in Section 2.2. Adding some more details there might help readers which are not already experts in OT to understand what is going on.
- The applied metrics in the experiments are reasonable but still make it hard to see in which way we actually deviate from the targeted true posterior. The authors could perhaps also show something like bias in posterior mean and SD or some other easy to understand quantities.
- Curse of dimensionality: These scaling issues are mentioned by the authors themselves and seem to be severe in the number of model parameters(?) I would like to see the scaling laws discussed in more detail here if possible.

**Questions:**

- I also don’t really see in Section 3, where exactly the real-world labels (the parameters underlying the real data) are actually used. Can you explain this?
- In EQ 3, you make a central assumption. Can you explain why you think this assumption is usually justified? Asked differently, when would it not be justified?
- In equation 6: I don’t fully understand the logic of why 6 is a valid approach to learn a summary network of the real data. Can you explain this to me in more detail?
- I don’t quite understand the results in relation to the LPP and ACAUC metrics:
    - So 0 would be optimal for ACAUC, right?
    - And for LPP higher values would be better, right?
    - If so, SBI looks better than ROPE. I know this is probably not how these metrics should be read, so I am merely asking to clarify this point for me.

---

> ### Author Response · Authors · 2024-11-14
> **Authors' response**
>
> We thank you for the very valuable feedback. We hope our answers below address the major concerns you have.
>
> **The main weakness ... consider recommending acceptance.**
>
> We would like to point you to the second paragraph in the introduction, which motivates the use of a calibration set and explains in what real-world situations we can expect to have labeled real-world observations, together with examples. As we wrote, for such real-world applications, where the result of the inference must be accurate, there will most likely be a labeled dataset so the practitioner can verify that the pipeline works and build trust in the result.
>
> **As an expert in SBI ... OT to understand what is going on.**
>
> Thank you for raising this point - we will further expand this section to make it easier for non-expert readers. Do you have any more specific suggestions of what background is missing for you to better understand this part?
>
> **The applied metrics in the experiments are reasonable ... SD or some other easy to understand quantities.**
> Once we acknowledge model misspecification, we lose access to a simulator of the “true” generative model of the real-world data and thus also to the “true” posterior distribution. For instance, for the Task E and F, relying on a limited set of real-world data, we do not have access to the “true” posterior. Thus we chose to rely on two complementary metrics that can be used in practical settings, have clear interpretations and are only optimal for the “true” posterior, if we would know it. These metrics, the ACAUC and LPP, highlight if the credible intervals cover the true parameter value at the designated frequency and the informativeness (i.e., how much uncertainty reduction do we get by using the learnt posterior distribution) to infer the parameter of the posterior respectively. To contextualise the LPP numbers, we provide the “SBI” and “prior” baselines that represent an upper bound and a lower bound on the performance of well-calibrated models, when the independence assumption made in equation 3 holds.
> Nevertheless, we appreciate the suggestion for synthetic benchmarks, for which it is possible to compute the mean and SD of “true” posterior distribution. We are planning to add those results as a table in the appendix.
>
> **Curse of dimensionality ... in more detail here if possible.**
>
> It appears non-trivial to make any formal statement on the scalability of the method as N, the number of parameters of interest, increases in general. We believe this will depend strongly on the exact simulator and real-world data considered. Nevertheless, we can reason about some corner cases.
> In case the parameters are really independent, the worst we can do is to apply the method N times and it should lead to reasonably good results (as a function of the degree of misspecification and C, the size of the calibration set). We could also apply RoPE jointly and hope to gain data efficiency as the sources of misspecification could be common to multiple parameters and the finetuning step would thus rely on strictly more information than if it only considered one parameter at a time. A toy example would be $\theta := \mu \in \mathbb{R}^k$ and $x_s \in \mathbb{R}^{k\times N} \sim \mathcal{N}(\mu, 1) \times \dots \times \mathcal{N}(\mu, 1)$ while   $x_o \in \mathbb{R}^{k\times N} \sim \mathcal{N}(\mu + c\mathbb{I}, 1) \times \dots \times \mathcal{N}(\mu + c\mathbb{I}, 1)$. This example highlights that, in certain cases, increasing the dimensionality of the parameters can help as this also implies that the calibration set is strictly more informative on the misspecification.
>
> Unfortunately, in other interesting cases there could be dependence between parameters (given the observation) that we would like to model as well as possible. In such cases, if these dependencies are poorly described by the simulator, we would expect the curse of dimensionality to hurt us and recovering these dependencies might necessitate exponentially larger calibration sets. Nevertheless, if the dependencies are well described by the simulator, we might again benefit from the more informative calibration set.

---

> > ### Author Response · Authors · 2024-11-14
> > **Authors' response [cont.]**
> >
> > **I also don’t  ... you explain this?**
> >
> > In section 3.1, we explain how we use the real-world calibration dataset, denoted $\mathcal{C}$ and composed of pairs of observations and labels (i.e., $\mathcal{C}:=\{(\theta^i, x_o^i)\}$ defined on line 128), to finetune the neural statistics. In particular the loss defined by equation (6) depends explicitly on $x_o^i$ and the $\theta^i$.
> >
> > **In EQ 3 ... when would it not be justified?**
> >
> > The assumption will likely not hold in many situations, e.g, in the real-world experiment in task E (the light tunnel). To see this, look closely at the synthetic and real images in task E, figure 2: in the synthetic data, the dimming effect of the polarizers affects only the colour of the hexagon. However, in the real data, the frames holding the polarizers (see [[1](https://arxiv.org/pdf/2404.11341), Figure 2] for a detailed diagram), introduce a reflection (the diffuse circular bands around the image edges), which carries information about the actual polarizer positions and can be used to infer the dimming effect $\alpha$. Thus, there is additional information about the parameters $\theta$ in the real observations $x_o$ that is not captured by the synthetic observations $x_s$, and thus the assumption in equation 3 does not hold. In other words, there is more signal in the real data than in the simulator. However, this does not prevent RoPE from working (see the results for task E in Figure 2) as it can still use the dependence modelled by the simulator to infer the value of the parameter. Moreover, as we elaborate in the paragraph below equation (3), this assumption can also have some benefits when the simulator encodes properties that the practitioner / domain expert believes are invariant across application environments and we want to avoid the predictive model to use dependence patterns (i.e., shortcuts) between the parameter and the observation that are not stable over other environments or distribution shifts.
> >
> > [1] Gamella, Juan L., Jonas Peters, and Peter Bühlmann. "The Causal Chambers: Real Physical Systems as a Testbed for AI Methodology." [arXiv preprint arXiv:2404.11341 (2024)](https://arxiv.org/pdf/2404.11341)
> >
> > **In equation 6 ... Can you explain this to me in more detail?**
> >
> > Thanks for raising this point, it seems we left a typo as $\mathcal{L}(\phi, \mathcal{X})$ should be $\mathcal{L}(\varphi, \mathcal{X})$. The loss enforces that the neural statistics on real data, $\mathbf{g}_{\varphi}(x_o)$ for a certain $\theta$ match with the expected neural statistics predicted for the simulation that corresponds to that $\theta$, $\mathbf{h}\_{\omega^\star} (S(\theta, \epsilon)$.
> >
> > **I don’t quite understand  the results in relation to the LPP and ACAUC metrics...**
> >
> > TL;DR: As we explain in line 387, the SBI baseline is an artificial (oracle) baseline that represents an upper bound on the LPP, i.e. the best performance that could be achieved if there was no misspecification. Therefore, it is just a reference and always performs better than the other methods—except MLP (in theory but not in our experiments), which does not use the simulator and directly exploits the signal in the real observations, see our point about EQ3 above.
> >
> > In more detail, here is how we compute the SBI baseline:
> > We take the ground-truth labels $\{(\theta^I\}\_{i=1}^N$ from the test set $\{(\theta^i,x^i_o)\}\_{i=1}^N$ on which we compute all the metrics for Figure 2; for each label $\theta^i$, we simulate an synthetic observation $x^i_s := \mathcal{S}(\theta^i)$, and collect them into a “synthetic” test set $\{(\theta^i,x^I\_s)\}\_{i=1}^N$; then, we apply to it the NSE+NPE pipeline (simulated posterior in Figure 1, right) to obtain the posterior estimates which we then evaluate. In this way, the baseline represents the performance we would hope to achieve if there was no misspecification and the simulator perfectly replicated the real observations (up to the stochasticity of the simulator itself).

---

> > ### Comment · Reviewer_MKFY · 2024-11-19
> >
> > Thank you for your response.
> >
> > 1) I understand your use cases where you argue for the existence of some labeled real data set. The application I work with (e.g., in psychology) have parameters (which would be the "labels" in your case) which are fundamentally unobservable. So where we would not get any real-world labled dataset. But I do see that they may be other use-cases.
> >
> > 2) The notation used in the definition of mathcal{B}_0 after the condition operator is unclear to me. As a result also the notation in EQ2 only makes sense half-way.
> >
> > 3) I see that of course we don't have a ground truth in the real world datasets. But what about the other datasets? There we could apply other (ground truth) metrics too. Thank you for planning to add them.
> >
> > 4) Thank you for your explanation on the curse of dimensionality. I see why it is hard to reason about it formally but what abut empirically? This could be studied in the experiments too, right?

---

> ### Author Response · Authors · 2024-11-20
> **Answer to follow up questions**
>
> Thank you for your follow-up questions. We have addressed them below and hope our responses provide the necessary clarifications. We believe the points raised can be effectively addressed through minor revisions to the current manuscript. However, if further clarification is needed, we would be happy to provide it.
>
> **I understand your use cases where you argue for the existence of some labeled real data set. The application I work with (e.g., in psychology) have parameters (which would be the "labels" in your case) which are fundamentally unobservable. So where we would not get any real-world labled dataset. But I do see that they may be other use-cases.**
>
> We absolutely agree that there are many applications of SBI where the inference parameters are unobservable. Our goal was to explore an alternative approach for the case when they are, because there are interesting real-world applications where this is the case where SBI could prove very valuable. Is this clear from the second paragraph in the introduction?
>
> **The notation used in the definition of mathcal{B}_0 after the condition operator is unclear to me. As a result also the notation in EQ2 only makes sense half-way.**
>
> Thank you for pointing this out, the notation is indeed a bit confusing. Here’s an alternative expression for line 179
> $$\mathcal{B}\_o = \{P\in\mathbb{R}^{N\_o\times N\_s}\_+ : \textstyle\sum\_{j=1}^{N\_s} P\_{ij} = \frac{1}{N\_o} \;\forall i=1,...,N\_o \}$$
>
> For equation 2 we will keep the notation but add an explanation, i.e.
>
> $$P^\star = \arg\min\_{P\in\mathcal{B}\_o} \langle P, C\,\rangle + \rho\mathop{KL}\left(P^T\mathbf{1}\_{N\_o} \| \frac{\mathbf{1}\_{N_s}}{N\_s}\right) + \gamma \langle P, \log P\rangle,$$
> where $\mathbf{1}\_{n}$ is a vector of ones with size $n$ and $\mathop{KL}\left(P^T\mathbf{1}\_{N\_o} \| \frac{\mathbf{1}\_{N\_s}}{N\_s}\right)$ is the Kullback-Leibler divergence between the marginal distribution over the simulated observations implied by the transport matrix $P$ and the uniform distribution (where each observation has the same probability).
>
> What do you think? Would this be clearer?
>
> **I see that of course we don't have a ground truth in the real world datasets. But what about the other datasets? There we could apply other (ground truth) metrics too. Thank you for planning to add them.**
>
> We also don’t have ground truth for synthetic benchmarks because we don’t have a closed-form expression for the posterior; we can approximate it using NPE (which is the posterior estimated by the “SBI” baseline in our experiments). If you still feel this is valuable, we can add (in the Appendix) some additional analysis (e.g., metrics & visualizations) using this proxy posterior.
>
> **Thank you for your explanation on the curse of dimensionality. I see why it is hard to reason about it formally but what abut empirically? This could be studied in the experiments too, right?**
>
> We fully agree with your point that developing algorithms capable of robustly predicting posterior distributions over high-dimensional parameter spaces, even in the presence of model misspecification, is a crucial research direction. We also acknowledge that this is not something RoPE explicitly addresses in its current form, nor has it been empirically demonstrated yet. We hope RoPE can be a good source of inspiration for paper tackling such settings. Nevertheless, we believe this falls outside the scope of our paper, which focuses on addressing SBI under model misspecification in its more common form—namely, for low-dimensional parameter spaces—an area explored by state-of-the-art papers [1-2].
> By introducing two real-world benchmarks and evaluating population-level metrics, we believe our work advances the experimental validation further than previous studies. Furthermore, we argue that any high-dimensional experiment should be grounded in a real-world problem, using realistic data to reflect genuine rather than idealized model misspecification. In such cases, evaluating the "correctness" of posterior distributions becomes inherently complex and application-dependent, requiring careful consideration of the specific scenario to ensure meaningful results.
> While we agree that this is an important and interesting challenge—relevant, for example, to climate and weather modeling or gravitational lensing—we believe it warrants a dedicated study in its own right. We also clearly acknowledge that applying RoPE to such setting may likely requires certain modification to the algorithm proposed.
>
> [1]:Huang, Daolang, et al. "Learning robust statistics for simulation-based inference under model misspecification." Advances in Neural Information Processing Systems 36 (2024).
>
> [2]:Ward, Daniel, et al. "Robust neural posterior estimation and statistical model criticism." Advances in Neural Information Processing Systems 35 (2022): 33845-33859.

---

> > ### Comment · Reviewer_MKFY · 2024-11-26
> >
> > Thank you again for your responses. I have raised my score from 5 to 6 now.

---

> > > ### Author Response · Authors · 2024-11-26
> > > **Thank you for the constructive feedback**
> > >
> > > We would like to thank you for your careful feedback and raising your score to 6 in light of this discussion, which was very constructive for us.

---

### Official Review · Reviewer_hywB · 2024-11-01

**Soundness:** 3
**Presentation:** 2
**Contribution:** 4
**Rating:** 8
**Confidence:** 4

**Summary:**

The paper introduces a method for addressing model misspecification in simulation-based
inference (SBI)  settings. Common SBI algorithms usually assume that the simulator (the
model) is well-specified, i.e., that it can simulate data close to the observed data.
However, if this is not the case, the resulting approximated posterior distributions can
be substantially biased. This paper proposes a procedure to mitigate this problem for
scenarios where multiple observed data points are available.

The authors propose to first follow a standard SBI approach of learning a neural-network
based embedding on the simulated data, followed by neural posterior estimation (NPE) on
the embedded data. Their key contributions consists of three additional steps. First,
they fine-tune the neural embedding using the set of observed data points. Second, they
then solve the optimal transport problem between the embedding simulated data and the
embedded observed data. Finally, at inference time, they use the optimal transport
solution between the embeddings to obtain the posterior given the observed data as a
weighted sum of the posteriors given simulated data.

The proposed method is evaluated on six benchmarking tasks for misspecification, two
existing tasks and four new tasks. It is compared to baselines and competing algorithms
and performs well in most settings. Additionally, the authors show results for ablations
and the method's behaviour for different hyperparameter settings.

**Strengths:**

The paper is well-written and well-structured. The first two sections
give a good introduction to the topic, accurately cite previous work and clearly state
the contributions and the specific SBI setting the method is designed to solve. The
paper uses two existing benchmarks and introduces four new model misspecification
benchmarks, which is a valuable contribution for the community. The performance of the
proposed method is promising and the ablation and hyperparameter sensitivity studies
help understanding the method and tend to make it easier for practitioners to choose
parameter in practice.

**Weaknesses:**

The new definition of model misspecification in SBI is a bit unclear
(see questions below) and should be clarified. Some of the benchmarking results of the
proposed method are unclear and concerning and need clarification. In general, the
figures in the experiments sections are difficult to read and make the overall results
unclear (see suggestions below).

**Initial recommendation:**

Overall, I believe this could be a valuable contribution for the field and I am
tentatively recommending to accept this paper. Model misspecification is an urgent
problem in SBI and methods for addressing it have been scarce so far. The proposed
method and additionally the proposed benchmarking tasks are a promising next step.
However, there are several questions that should be clarified and several improvements
to the visualizations of the results that should be made. The performace of RoPE on
several benchmarking tasks seems problematic to me, but this might be due to a
misunderstanding of the ACAUC metric and needs clarification (see question 11 below).

**Questions:**

1) Definition of misspecification in SBI: There are several parts in the definition that
I find challenging to follow. In line 090 it says

> our simulator models the relationship between the real observations xo and the
> parameters of interest as they appear in the calibration set

and for that sentence it is followed, that the current definition is insufficient
because a model may be well-specified by not well-calibrated. I cannot follow this line
of argument. Are you defining mispecification based on posterior calibration here? Can
you please clarify? (I have seen the example in Appendix B, but it shown only that the
posterior is biased, not badly calibrated).

2) Related to my first question, I am wondering how your definition differs from the one
   given in https://arxiv.org/abs/1904.04551, who introduced a synthetic likelihood
   approach for misspecified simulators.

3) In line 201, $\pi$ is introduced as a joint distribution. However, below in equation
   (4) it is then used as conditional distribution. Can you clarify this change please?

4) In line 257, RoPE$^*$ is defined with $\tau < 1$, but how much smaller than 1 does it
   have to be? Further below, it is set to $\tau=0.9$. How is chosen in practice?

5) In line 409, it is claimed that RoPE significantly reduces the uncertainty compared
   to the prior. How is this measured? Is LPP your proxy for this? If so, please clarify
   in the text.

6) How is ACAUC related to the more common posterior calibration metrics like SBC or
   expected coverage? In line 381, it says that it is zero for a perfectly specified
   prior. Shouldn't it refer to the calibration performance of the posterior instead?

7) The **SBI** defined in line 387 is unclear to me. How is it obtained precisely and
   how can it be the upper bound on the performance in misspecified simulators? How can
   it below the prior baseline in the ACAUC plots for some tasks?

8) How can the MLP baseline work so well when it assumes a Gaussian posterior? Are the
   benchmarking task posteriors all close to Gaussian?

9) The choice of $\gamma$ can have a substantial effect on the accuracy of the
   posterior, tending towards the prior for too large values of $\gamma$. This would be
   highly undesirable in practice. How can the practitioner tune $\gamma$ for real-world
   applications, i.e., outside the benchmarking tasks where it seems clear what the upper
   and lower bound of the metric are?

10) General remark on the figures: Figure 1 is great, but figure 2-4 are packed quite
    challenging to read. The dotted lines are barely visible and quite different than
    shown in the legend. Some lines have markers, others do not. Especially in the ACAUC
    plots it is almost impossible to tell apart the different lines. Smaller line width
    would be a start, but a general improvement in the choice of color code and markers
    would be even better.

11) Some questions on the RoPE performance: Why does it perform so badly compared to NPE
    on the SIR example? You mentioned that the misspecification in SIR is weaker. How
    does RoPE perform on well-specified simulators in general then? Could this be a
    general weakness of the method?
    Also, RoPE seems to be generally underconfident in terms of ACAUC and this does not
    improve with calibration set size. This is concerning as well. I think it would be
    important to check and compare the posterior predictive distributions as well.


**Additional feedback**

- The reference to figures from the main text is a bit off. For example, figure 1
  appears on page 2, but is referenced only on page 5. Figures 3 and 4 are referenced
  before figure 2.
- A lot of content is moved to the appendix. It would be better to move parts of it back
  to main text, e.g., algorithm 1. Or give more context in the main text. For example,
  to me it became clear only by looking at the algoritm, that at inference time, one
  needs to run additional simulations to match the test set of observed data for
  computing the weighted posterior.

---

> ### Comment · Reviewer_hywB · 2024-11-22
>
> Dear authors,
>
> The rebuttal period is almost over and you have not yet responded to my review and the open questions. While some of my questions have been answered in the other responses, I would still appreciate a response.
>
> Best wishes,
> Your reviewer

---

> > ### Author Response · Authors · 2024-11-22
> > **Response to your questions**
> >
> > Apologies for the delay, we sincerely appreciated your feedback and especially the fact that you positively assessed the relevance and soundness of our work. We now to provide a complete response to your questions below. Please let us know if you have any follow up questions.
> >
> > **Q1 & Q2:**
> >
> > After our discussion with reviewer yfyx, we have decided to update our definition with an equivalent formulation, which we believe is clearer:
> >
> > [New definition start]
> >
> > First, we assume the calibration set $\{(\theta^i, \mathbf{x}\_o^i)\}\_{i=1}^{N\_c}$ of real-world observations $\mathbf{x}\_o \in \mathcal{X}$ and their corresponding labels $\theta \in \Theta$ are sampled i.i.d.\ from a joint distribution given by the density $p^\star(\theta,\mathbf{x}\_o)$. Let $p^\star(\theta)$ be the marginal density of the underlying parameters $\theta$ in the real world, and $p^\star(\mathbf{x}\_o) := \int_\Theta p^\star(\theta) p^\star(\mathbf{x}\_o \mid \theta) d\theta$ be the marginal density of the real-world observations, where $p^\star(\mathbf{x}\_o \mid \theta)$ is the unknown process which is modeled by the simulator $p(\mathbf{x}\_s\mid \theta)$. We say the simulator is misspecified if $\exists \mathcal{S} \subseteq \Theta\times\mathcal{X}$ with $\iint\_\mathcal{S} p^\star(\theta, \mathbf{x})d\theta d\mathbf{x} > 0$ such that $p^\star(\mathbf{x}\_o\mid\theta) \neq p(\mathbf{x}\_s=\mathbf{x}\_o\mid\theta)$.
> >
> > [New definition end]
> >
> > We hope this new definition avoids the confusion produced by basing it on posterior "correctness" (which goes beyond calibration as for instance the prior is a calibrated posterior but not necessarily the "correct" posterior). Note that it is still equivalent when we define the quantities
> >
> > $
> > p^\star(\theta \mid \mathbf{x}_o) := \frac{p^\star(\theta)p^\star(\mathbf{x}_o \mid \theta)}{p^\star(\mathbf{x}_o)}
> > $
> > and
> > $
> > p(\theta \mid \mathbf{x}\_o):= \frac{p^\star(\theta) p(\mathbf{x}\_s = \mathbf{x}\_o \mid \theta)}{p^\star(\mathbf{x}\_o)}.$
> >
> > In light of this reformulation, note that our definition is stronger than the one referenced in lines 80-90): even when the true data generating process does belong in the family of distributions induced by the simulator, the simulator may still be misspecified under our definition because we require that for a fixed value of $\theta$---instead of “some” value of $\theta$---the simulator matches the distribution of the real data. This is what we show in the example in Appendix B. We chose a simple example where the posterior is biased—and thus provides the wrong coverage / credible intervals, i.e., it is not well calibrated.
> >
> > This is also the main difference with the reference you provide (question 2). A more concrete example of this is the color shift observed in the light tunnel experiment, where the pixel color in real-world measurements does not perfectly correspond to the RGB values set on the actuator to produce that color. If we defined the observation as the center pixel of the image, our simulator—which can generate any pixel color— would be able to replicate all measured observations. However, the color-shift effect, which is difficult to model, would still lead to a wrong inference of the underlying parameters. In contrast, the definition of model misspecification used in the synthetic likelihood approach would not classify this situation as misspecified.
> >
> > **Q3:**
> >
> > We implicitly used the same notation to indicate that this is the conditional distribution corresponding to that joint distribution, i.e., $\pi^\star(x\_s \mid x\_o) = \frac{\pi^\star(x\_s, x\_o)}{\int \pi^\star(x\_s, x\_o) dx\_s}.$
> >
> > **Q4:**
> >
> > $\text{RoPE}^\star$ is introduced as an alternative to RoPE when the practitioner is not confident that the prior distribution $p(\theta)$ accurately describes the marginal distribution of $\theta$ behind the test set. We see $\tau$ as an hyperparameter that allows the RoPE to move away for the prescribed prior distribution and its optimal value will thus be problem dependent. We observed that $\tau = 0.9$ provided a good tradeoff in our experiments, showing robustness to prior misspecification but also working well when the prior is well-specified. We study further the effect of Tau in Figure 4. However, as any hyperparameter its value can be optimized depending on some validation metrics, which remain dependent on the targeted application.

---

> > > ### Author Response · Authors · 2024-11-22
> > > **Response to your questions [cont.]**
> > >
> > > **Q5:**
> > >
> > > Yes, LPP is a proxy for this, particularly the gap between the LPP of the prior and the LPP of the posterior. In lines 370 to 373, we explain why this is the case:
> > > “The LPP is an empirical estimation of the expectation over possible observations of the negative cross-entropy between the true and estimated posterior; thus, for an infinite test set, it is only maximized by the true posterior. LPP characterizes the entropy reduction in the estimation of $\theta$ achieved by a posterior estimator $\tilde{p}$ when given one observation, on average, over the test set.”
> > >
> > > Does this explanation, which relates LPP to cross-entropy and the modeled mutual information between the parameter and the observation, provide sufficient intuition?
> > >
> > > **Q6:**
> > >
> > > The ACAUC is closely related to SBC (simulation-based calibration) or expected coverage, as it corresponds to the integral over credibility levels $\alpha$ of the difference between $\alpha$ and the empirically observed frequency of samples $\theta$ that fall within the posterior distribution at that level $\alpha$. Visually, if we plot the empirical coverage as a function of the level $\alpha$, the ACAUC corresponds to the integral of the signed area between this curve and the diagonal, the latter representing the function we would observe for a perfectly calibrated posterior.
> > >
> > > A limitation of this metric is that if the posteriors are, on average, overconfident for certain levels of $\alpha$ and underconfident for others, the ACAUC may still be small even though the posterior is not perfectly calibrated. The ACAUC can also be interpreted as a measure of whether, by uniformly choosing confidence levels, we will obtain, on average, underconfident or overconfident credible intervals.
> > >
> > > Regarding line 381: no, we are simply reminding the reader that the prior distribution has a coverage curve that lies on the diagonal and therefore results in an ACAUC of zero.
> > >
> > > **Q7:**
> > >
> > > TL;DR (more details below): It is a neural posterior estimator trained and tested on simulated data. The LPP and ACAUC metrics we report for this baseline are computed on simulated data. Thus, the LPP represents an upper bound on RoPE’s performance because of the independence assumption made between x_o and theta given x_s –  x_s acts as an information bottleneck in RoPE. If the NPE returned a perfect estimate of the posterior distribution, the ACAUC metric should be equal to 0. However, as shown by [1], obtaining perfectly calibrated SBI models (even without considering mode misspecification) is difficult, which can result in the estimate being underconfident (and thus having a negative ACAUC score).
> > >
> > > In more detail, here is how we compute the SBI baseline:
> > >
> > > We take the ground-truth labels $\{(\theta^i\}\_{i=1}^N$ from the test set $\{(\theta^i,x^i\_o)\}_{i=1}^N$ on which we compute all the metrics for Figure 2; for each label $\theta^i$, we simulate an synthetic observation $x^i\_s := \mathcal{S}(\theta^i)$, and collect them into a “synthetic” test set $\{(\theta^i,x^i\_s)\}\_{i=1}^N$; then, we apply to it the NSE+NPE pipeline (simulated posterior in Figure 1, right) to obtain the posterior estimates which we then evaluate. In this way, the baseline represents the performance we would hope to achieve if there was no misspecification and the simulator perfectly replicated the real observations (up to the stochasticity of the simulator itself).
> > >
> > > [1]:Hermans, Joeri, et al. "A crisis in simulation-based inference? beware, your posterior approximations can be unfaithful." Transactions on Machine Learning Research (2022).
> > >
> > > **Q8:**
> > > Indeed, the performance of the MLP—in terms of LPP—increases significantly with the size of the calibration set. This is because it extracts all possible information from the observations (even beyond the simulator) to essentially solve a supervised prediction problem, and creates a sharp (gaussian) posterior around the true value. Thus, the true value is often assigned a high likelihood where the model generalises well, hence the high LLP scores with large calibration sets.
> > >
> > > However, if you evaluate the calibration of the estimated posterior with the ACAUC score, the performance is much worse, even at large calibration set sizes, except for tasks C and E, where the true posterior seems to be well represented by a unimodal/unskewed distribution.
> > >
> > > In any case, if the calibration set size is so large as to allow performance of a supervised method, then that would likely preclude the need to use SBI. We include the MLP baseline and the large calibration set sizes to illustrate how, by using SBI that is robust to misspecification, we can leverage a rough simulator to drastically reduce the amount of samples needed for an accurate estimator. We consider this one of the most exciting take-aways of the paper.

---

> > > > ### Author Response · Authors · 2024-11-22
> > > > **Response to your questions [cont.]**
> > > >
> > > > **Q9:**
> > > >
> > > > We provide a discussion indications to set $\gamma$ for the practitioner in lines 473-480. Please let us know what you think.
> > > >
> > > >
> > > > **Q10:**
> > > >
> > > > Thank you for your feedback, we will adjust the figures.
> > > >
> > > > **Q11:**
> > > >
> > > > We appreciate this important point and would like to clarify that we deliberately avoided over-optimizing the performance of RoPE–i.e., its hyperparameters–for each benchmark. Despite this, RoPE consistently provided informative posteriors while avoiding overconfidence—a highly desirable property. When RoPE indicates that certain parameter values are unlikely, these predictions can be trusted.
> > > >
> > > > The results reported in Figure 2 are for $\gamma = 0.5$, although we observed better performance with $\gamma = 0.1$ on certain benchmarks, particularly for the SIR benchmark, where $\gamma = 0.5$ leads to the underconfident results you mention. The observed underconfidence arises from the fine-tuning step potentially overfitting in the absence of misspecification, compounded by the cautious predictions resulting from entropic optimal transport caused by $\gamma=0.5$. We intentionally chose not to optimize $\gamma$ for individual benchmarks to maintain fairness and consistency in our evaluations.
> > > > Additionally, we acknowledge that SIR may not be the most ideal benchmark for evaluating robust SBI methods, as standard NPE already performs well on this task, achieving results close to those obtained if the simulator is well-specified. In such settings, robust SBI methods may not be strictly necessary. That said, we included the SIR benchmark for completeness, as it has been widely used in prior work on SBI under misspecification.
> > > >
> > > > **Additional feedback:**
> > > >
> > > > Re (figures): Thanks for noticing that. We will move Figure 1 to page 5. However, Figure 2 contains the main results and we believe it should appear earlier in the text, as it gives context for Figure 3 and 4 that show additional results (even if they are mentioned earlier).
> > > >
> > > > Re (appendix): We had to relegate some context to the appendix due to space constraints. An option is to substitute Fig. 1 by Algorithm 1, but we felt the visualizations of the problem setup and the algorithm components were more illustrative. We would like to note that it is not necessary to generate new simulation samples at inference time as we can simply re-use the ones used to train the NPE model.

---

> > > > > ### Comment · Reviewer_hywB · 2024-11-22
> > > > >
> > > > > Thank you for the prompt and extensive responses to my concerns and open questions. There are all resolved now, or will hopefully be addressed by you in an updated version of the manuscript.
> > > > >
> > > > > In particular, your explanations and clarifications for questions 7, 8 and 11 should find their way into the main text.
> > > > >
> > > > > Overall, with the updated definition of model misspecification and the clearer communication that the use-case you are addressing here requires a (small) labelled real-world data set, I believe this is a good paper and I will adapt my score accordingly.

---

> > > > > > ### Author Response · Authors · 2024-11-25
> > > > > > **Thanks for the constructive feedback!**
> > > > > >
> > > > > > We would like to sincerely thank you for carefully considering our work, providing valuable feedback and for increasing your score in the regards of our discussion.

---

### Official Review · Reviewer_6VTN · 2024-11-03

**Soundness:** 3
**Presentation:** 2
**Contribution:** 2
**Rating:** 5
**Confidence:** 4

**Summary:**

This work contributes another contender to the family of “robust” methods for neural SBI, which is known to be unreliable for out-of-distribution (OOD) data. The basic idea is to use a “calibration set” of observables and ground-truth measurements as an additional training signal to inform the neural network responsible for embedding the observables into a lower-dimensional representation about distribution shifts and reduce its susceptibility to OOD observables. During inference, an optimal transport (OT) cost matrix is used to derive weights for approximating the posterior over observables as a weighted average over an ensemble of posteriors obtained from the training simulations.

**Strengths:**

- The paper contributes an interesting approach to a hard problem in SBI and scientific modeling in general.
- The OT formulation and ensemble-based correction are well motivated and the assumptions are clearly stated.
- The evaluation in the setting of low-dimensional models is comprehensive and features good comparisons. The results are encouraging even with a very small calibration set.

**Weaknesses:**

- The basic premise of the work is that “ground truth” parameters or latent quantities are available during training. In other words, it assumes that the simulator is parameterized in the same way as reality. This severely limits the applicability of the proposed method beyond certain domains of physics physics, as “true” latent quantities are not available or not even sought-for in a majority of applications of Bayesian inference outside of physics. Nevertheless, the authors openly acknowledge this limitation and I don’t think that it is an unsurmountable dealbreaker. On a related note, this limitation extends to the formal definition of misspecification assuming there is a “true prior”, and correspondingly, a “true posterior” that represents the updated prior which is hardly actionable - the marginal likelihood $p(x_0)$ is a model-implied quantity and de Finetti’s representation theorem about the existence of a factorization apples only to infinitely exchangeable sequences without any guarantees that $dim(\theta)$ is finite.
- The authors are quite selective in their choice of cited related work. For instance, the authors can find extensive discussions on model misspecification in likelihood-free inference in [1, 2, 3, 4, see also references therein]; [5] also discuss an alternative definition of model misspecification through the scope of the marginal likelihood, which is also explicitly discussed in [6]. [7] propose to view misspecification as latent-space distortion and correct it during inference using the reverse KL. This goes on to show that the proposed paper can profit from a slightly more in-depth discussion on conceptual and empirical approaches to dealing with model misspecification and perhaps offer a categorization of methods designed to *detect* vs. methods designed to *fix* potential errors. Additionally, reference [8] regarding universal density approximation is misleading; see [9 and references therein] for a comprehensive treatment of the university of coupling-based normalizing flows.
- I have two concerns regarding the scalability of the method. First, it seems that the transport matrix can grow very large for settings with millions of observations and millions of simulations. This, coupled with the need to optimize the optimize the $\gamma$ trade-off parameter makes the method attractive only for small-data, small-simulation-budget settings. Second, the empirical evaluation only considers models with extremely low parameter dimensionality $\theta$ (i.e., the largest parameter space has 4 parameters), which is a serious limitation and leaves the question of generalizability to more complex models completely open. As such, I believe the claims of the paper to be a bit over the top. I do appreciate that the evaluations were performed on fairly large tests sets of 2000 simulations and that recent contenders and good ablations were considered.
- The presentation of the results is rather hard to follow. I believe some of the model details (e.g., example data sets) could have been delegated to the appendix and sushi plots summarized more succinctly. I am a bit concerned by the lack of sensitivity regarding the size of the calibration set (apart from task E).

[1] Nott, D. J., Drovandi, C., & Frazier, D. T. (2023). Bayesian inference for misspecified generative models. Annual Review of Statistics and Its Application, 11.

[2] Frazier, D. T., Robert, C. P., & Rousseau, J. (2020). Model misspecification in approximate Bayesian computation: consequences and diagnostics. Journal of the Royal Statistical Society Series B: Statistical Methodology, 82(2), 421-444.

[3] Frazier, D. T., Kohn, R., Drovandi, C., & Gunawan, D. (2023). Reliable Bayesian inference in misspecified models. arXiv preprint arXiv:2302.06031.

[4] Kelly, R., Nott, D. J., Frazier, D. T., Warne, D., & Drovandi, C. (2024). Misspecification-robust sequential neural likelihood for simulation-based inference. Transactions on Machine Learning Research.

[5] Schmitt, M., Bürkner, P. C., Köthe, U., & Radev, S. T. (2023, September). Detecting model misspecification in amortized Bayesian inference with neural networks. In DAGM German Conference on Pattern Recognition.

[6] Masegosa, A. (2020). Learning under model misspecification: Applications to variational and ensemble methods. Advances in Neural Information Processing Systems, 33, 5479-5491.

[7] Siahkoohi, A., Rizzuti, G., Orozco, R., & Herrmann, F. J. (2023). Reliable amortized variational inference with physics-based latent distribution correction. Geophysics, 88(3), R297-R322.

[8] Wehenkel, A., & Louppe, G. (2019). Unconstrained monotonic neural networks. Advances in neural information processing systems, 32.

[9] Draxler, F., Wahl, S., Schnörr, C., & Köthe, U. (2024). On the universality of coupling-based normalizing flows. arXiv preprint arXiv:2402.06578.

**Questions:**

Can the authors explain the insensitivity of performance to the size of the calibration set?
Can the method be extended to incorporate a less-than-perfect calibration set (e.g., as coming from a well-calibrated posterior estimator from another data set)?

---

> ### Author Response · Authors · 2024-11-18
> **Authors' response**
>
> We are sincerely thankful for your review and we understand some of your concerns. We hope our discussion below will convince you that those concerns can be addressed in the scope of small revisions. We also believe, that some of your criticism is subjective and does not directly concern our contribution but the types of real-world problems we versus you are interested in solving (see lines 44-78). We hope to find common ground where our contribution can be seen as a sound and original step towards better solutions to address model misspecification in the context of simulation-based inference.
>
> **The basic premise of the work ... without any guarantees that  is finite.**
>
> We agree with you there are settings–such as the ones you describe–for which our solution would not be of direct use. This is why we explicitly acknowledge this limitation in the second paragraph. Nevertheless, SBI is also often used in a context where 1) we have access to a model in the form of a simulator, 2) the model’s parameters exist beyond the scope of the considered model (e.g., they represent physical quantities), and 3) the goal is to infer their value given the simulated measurement. We also agree that the “true prior” and “posterior” are artificial objects that are, again, a model of the real world. Nevertheless, we use these objects to represent an oracle solution that would be optimal, which represents the best inference we would be able to do if we had infinite access to the real-world generative process. We believe it is important to make a certain distinction between the “prior” and the “likelihood model” in our context as 1) these two objects are instantiated very differently in the context of SBI (one is just a marginal distribution over parameters, the other is a black-box computer program) and 2) the parameter $\theta$ and measurements $x$ have very concrete interpretations. The (true) prior defines the marginal distribution of parameters in the real world, which can be estimated, e.g., by a large set of labels, but does not necessarily require paired measurement. In contrast, the likelihood model is represented by the computer program (as a stochastic map from parameter to observations).
>
> **The authors are quite selective ... coupling-based normalizing flows.**
>
> We agree [1-7] are relevant work, as they also discuss and aim to address the issue of model misspecification, that are worth being discussed in the scope of our paper. However, we would like to emphasize that in many situations where SBI is relevant, the parameters are interpretable (e.g., these parameters are given physical units). It may be expected that the practitioner would like a solution to model misspecification that does not hurt this interpretation. In our paper, we suggest that it is possible to keep this interpretability with a calibration set where this interpretability is respected. In contrast, we believe the solutions proposed in these works are not viable in the case where the model’s parameters are of direct interest, as they do enforce in any way that this interpretation is kept. We are doubtful there exists a non-data-driven way of addressing model misspecification when keeping the parameter interpretation as needed in the settings we define in lines 44-78. We will add a discussion on these papers and believe this could indeed help the reader to better understand our contribution.
>
> While we agree it is worth adding [9], we do not believe reference [8] is misleading. It introduces monotonic transformations parameterized by neural networks and demonstrates that these networks are universal approximators of continuous monotonic functions. Following this, authors can conclude that autoregressive normalizing flows that rely on these transformations can represent arbitrarily expressive classes of continuous density approximators (see “Universality” paragraph before section 4). To the best of our knowledge, this is the first work to claim that NF are universal density estimators of continuous random variables. We also rely on autoregressive NF, so results about coupling-based NF are less relevant.

---

> ### Author Response · Authors · 2024-11-18
> **Authors' response [cont.]**
>
> **I have two concerns ... good ablations were considered.**
>
> We appreciate your concern about scalability. However, our current focus is on settings where the test set size does not consist of “millions” of observations, although we agree this could be an interesting follow-up work. Optimal Transport is a very active research area and we believe adapting this part of RoPE to more specific setups, such as for very large test sets, should be valuable. For instance, in Appendix H, we briefly discuss how RoPE could be extended to handle larger sets of observations. This goes beyond the scope of our paper which rather aims to provide a simple and robust framework to handle model misspecification and to demonstrate that this strategy significantly outperforms others on a set of representative benchmarks.
>
> We use the simulated samples to (1) train the NPE, and (2) have a pool of samples to perform the OT matching during inference. Regarding (1), there is no more constraint than in the standard SBI settings there, as we aim to use as many simulations as possible to learn a posterior distribution that approximates the one implicitly defined by the simulator and prior distribution as well as possible. Regarding (2), in our experiments we observed that using a simulation set of similar size as the test set yielded good results on the benchmarks considered. Nevertheless, we agree that there might be marginal gains by increasing the size of that simulated dataset and that would increase the computational complexity of running RoPE. In that case, exploring a version of RoPE based on amortized OT transport is a promising avenue.
>
> Regarding the dimensionality of the considered model we would like to clarify two points. First, our empirical validation is significantly more extended than concurrent work (e.g., [4] and [5]) aiming to solve the issue of model misspecification in SBI, which typically consists of synthetic settings with one or two target parameters. Second, as it is common in SBI, we explicitly mention the parameters of interest, but the simulator (e.g. task F) often depends on a much larger set of nuisance parameters that are not of direct interest but impact the simulations and add complexity to the modeled generative process. We would like to emphasize that there exist many applications where the goal is to accurately infer a low dimensional parameter while quantifying uncertainty. For instance, when assessing the cardiovascular health of a subject, doctors only check a few vital parameters (e.g, central blood pressure, arterial stiffness or cardiac output, see related work in introduction). We believe that providing tools to create new measurement techniques to infer such low-dimensional quantities, from imperfect models and labeled calibration sets, is of great value for many domains of science and engineering. This low dimensionality also aligns well with the most prominent applications of SBI where, while the forward model is often very complex, the parameters of interest are much lower dimensional (e.g, neuronal activity tracking [1], Higgs discovery [2], or Gravitational analysis [3])
>
> We hope this addresses your concerns, but please let us know which claims you feel are still over the top.
>
> [1]Gonçalves, Pedro J., et al. "Training deep neural density estimators to identify mechanistic models of neural dynamics." Elife 9 (2020): e56261.
>
> [2]Brehmer, Johann, et al. "Mining gold from implicit models to improve likelihood-free inference." Proceedings of the National Academy of Sciences 117.10 (2020): 5242-5249.
>
> [3]Dax, Maximilian, et al. "Real-time gravitational wave science with neural posterior estimation." Physical review letters127.24 (2021): 241103.
>
> [4]:Huang, Daolang, et al. "Learning robust statistics for simulation-based inference under model misspecification." Advances in Neural Information Processing Systems 36 (2024).
>
> [5]:Ward, Daniel, et al. "Robust neural posterior estimation and statistical model criticism." Advances in Neural Information Processing Systems 35 (2022): 33845-33859.

---

> ### Author Response · Authors · 2024-11-18
> **Authors' response [cont.]**
>
> **The presentation of the results ... from task E).**
>
> We appreciate your feedback but believe the results section may be hard to follow because our evaluation is extremely complete and we try to set high standards for future works aiming to address model misspecification in SBI. We aimed to present our results objectively by contextualizing with several contenders, ablations and baselines. We believe this picture, while dense in information, also clearly highlights that RoPE systematically outperforms alternative strategies, providing informative posterior distributions that are well-calibrated. We also believe plotting the difference between simulated and real-world data is important as it provides some context to the reader to understand that we do not only consider “toy” or “naive” misspecification. Do you have any specific advice to make our results easier to read without losing information?
>
> What do you mean by “sushi” plot?
>
> **Can the authors explain the insensitivity of performance to the size of the calibration set?**
>
> We believe these results demonstrate that combining a supervised fine-tuning step together with an OT step, which ensures the observed and simulated marginal distributions of the data are matched to each other, is an effective way to exploit both supervised and unsupervised data. Nevertheless, RoPE is not perfect and there may be multiple reasons for the performance to quickly saturate. Indeed, if the simulator is extremely good or bad, it may respectively make the issue of misspecification very easy or impossible to address by RoPE. In this case, we would observe a plateau. In addition, the RoPE fine-tuning strategy is simple and may be suboptimal in cases where the calibration set size is larger, explaining why performance plateaus on some benchmarks. Nevertheless, RoPE systematically outperforms baselines, yielding well-calibrated and informative posterior distributions. The RoPE framework could be improved for particular use cases by adapting the fine-tuning step and ensuring it extracts as much information as possible from the real data, while aligning them with the sufficient statistics from the simulated data.
>
> **Can the method be extended to incorporate a less-than-perfect calibration set (e.g., as coming from a well-calibrated posterior estimator from another data set)?**
>
> Yes, we believe that exploring different fine-tuning strategies, tailored to specific scenarios, are interesting future directions. For instance, the current loss could be adapted to use samples from the well-calibrated posterior instead of exact values. A potentially reasonable fine-tuning loss could be:
> $$\mathcal{L}(\varphi;\mathcal{C}, p^\star) := \sum\_{i=1}^{N_c} \lvert \mathbf{g}\_{\varphi}(\mathbf{x}\_o^i) - \mathbb{E}\_{\theta \sim p^\star(\theta \mid (\mathbf{x}\_o^i)}\mathbb{E}\_{\varepsilon \sim \mathcal{U}[0, 1]}[ \mathbf{h}\_{\omega^\star}\left(S(\theta, \varepsilon)\right) ] \rvert\_2,  $$, where $\mathcal{C}$ only contains real-world observations and $p^\star$ is this well-calibrated posterior.

---

> > ### Comment · Reviewer_6VTN · 2024-11-26
> >
> > I have read the other reviews and thank the authors for their responses and clarifications. While I agree that the closed world setting including "true priors" and "true parameters" is indeed an interesting one in certain disciplines, my concerns about the scalability and scope of the method remain. In particular, my enthusiasm is tampered down, as I currently still only see marginal benefits for small-data, small-simulation-budget, low-parameter dimensions settings. I remain convinced that the experimental evaluation can be greatly diversified by qualitatively different properties (e.g., low. vs. high dimensional settings) instead of quantity. I have increased my score from 3 to 5 to acknowledge the author's engagement with all reviewer feedback.

---

> > > ### Author Response · Authors · 2024-11-27
> > > **Thank you for the feedback**
> > >
> > > We sincerely appreciate your thoughtful engagement with our work and for increasing your score from 3 to 5. In particular, we believe that your feedback has directly contributed to improving the paper.
> > >
> > > We acknowledge and share your perspective on the importance of diversifying experimental evaluations to explore settings with qualitatively different properties, such as high-dimensional parameter spaces and varying numbers of (unlabelled) observations. While we believe our paper already raises the bar compared to prior work on SBI under model misspecification, we agree that extending benchmarks to cover such scenarios would provide valuable insights into the strengths and limitations of existing methods and is a crucial direction for future research.
> > >
> > > Thank you once again for your constructive input, which helped us strengthen our work.

---

### Official Review · Reviewer_YfYx · 2024-11-04

**Soundness:** 2
**Presentation:** 2
**Contribution:** 2
**Rating:** 5
**Confidence:** 4

**Summary:**

The paper presents a robust posterior estimation method (RoPE) for simulation-based inference (SBI) when the model is misspecified but the the underlying parameters of interest can be physically measured via (possibly expensive) experiments. RoPE uses such a calibration dataset to better quantify the posterior uncertainty under model misspecification scenarios.

**Strengths:**

The paper addresses a relevant practical problem in the field of SBI. The paper is largely well-written, and the experimental evaluation seems thorough. The idea of using the optimal transport coupling for learning statistics is neat (approach is similar to the work of Huang et al. (2023) who also learn statistics to be robust to model misspecification, without the calibration set of course).

**Weaknesses:**

I have concerns regarding the soundness of the proposed method. The main claim is that RoPE provides well-calibrated uncertainty quantification, which I am not convinced of. The self-calibration property discussed in Appendix F is under the assumption that the NPE posterio is well-calibrated, which itself is not guaranteed and is often overconfident as pointed out by [1] (and therefore making NPE well-calibrated is an active area of research in SBI [2]). Moreover, the expectation in equation 10 in the proof is estimated using $N_o$ Monte Carlo samples from the test set, which I am assuming, would be limited (unless having a large test set is one of the assumptions/requirements of RoPE, in which case it should be mentioned clearly since the other methods do not have such a requirement). Basically I do not see why RoPE solves the problem being addressed. Please let me know if I am missing something.

[1] Hermans et al. (2022). A Crisis In Simulation-Based Inference? Beware, Your Posterior Approximations Can Be Unfaithful. TMLR.
[2] Falkiewicz et al. (2023). Calibrating Neural Simulation-Based Inference with Differentiable Coverage Probability. NeurIPS.

Looking at how the authors define the model misspecification problem, it seems like the problem is of posterior calibration. Calling it "model misspecification" only makes it confusing since the definition is in terms of posteriors, and is not necessarily a property of the model (the example in Appendix A focuses on posterior bias rather than calibration). I do not see what is gained by calling this a "model misspecification problem", as a well-specified model may still produce mis-calibrated posteriors (mentioned in lines 92-93). I really think that the story becomes straightforward if the problem being addressed is of obtaining calibrated SBI posteriors (i.e. a "calibration problem"), both in cases when the model is well- or mis-specified (right?), which can be done using some additional information (in the form of calibration dataset), which is available in scenarios such as the cardiovascular system and industrial process. If the authors agree with this statement, then the more appropriate baseline for RoPE is the method of [2], instead of NPE-RS and NNPE.

The equation in line 190 is the same as equation 3 from the Ward et al. (2023) paper (NNPE method). Seems like the primary difference between the two is that RoPE uses OT to learn $p(\mathbf x_s | \mathbf x_o)$, while NNPE uses a spike and slab model. Please elaborate on the key differences between the two approaches, as it would help gauge the originality of this work. Also, it is not clear to me what is the role of the assumption in equation 3 then, as NNPE does not seem to have that assumption but still arrives at the same equation (correct me if I am wrong). I suppose lines 193-194 states why this assumption is needed, but it is not clear to me. Plus, there needs to be more discussion on how limiting is this assumption, in which cases is it satisfied and not satisfied, and the implications for when it does not hold.

The notation is unclear at times. For instance, what is the difference between $\mathbb{P}^\star$ and $p^\star(\mathbf x_o)$? Sometimes the author say $p^\star(\theta)$ is the true distribution of parameters in the real-world, but then they use $p^\star(\theta | \mathbf x_o)$ which is not introduced.

In section 2.2, it would be great if you could explain what an OT coupling is and what is the general idea (in simple words) before going into the OT details. This would help the readers not familiar with OT to get a sense of what this does without needing to understand every detail.

Minor comments:
* In equation 6, I think there should be $\varphi$ in the left-hand side instead of $\phi$, right?
* typo in line 817: "i.d.d." instead of "i.i.d".

**Questions:**

In section 3.1, the authors say that "it is reasonable to learn a sufficient statistic $\mathbf h_o$ by fine-tuning $\mathbf h_{\omega^\star}$". How do we know that such a fine-tuning will produce sufficient statistics? Are there any guarantees?

In most of the experiments, the performance of RoPE does not var as the calibration set size increases. Is there a reason why the performance of RoPE saturates in those cases even with $N_c = 10$? Are these very simple problems?

---

> ### Author Response · Authors · 2024-11-15
> **Authors' response**
>
> We are thankful for your careful feedback. We provide below clarifications and address the concerns you raised.
>
> **I have concerns ...  Please let me know if I am missing something.**
>
> We would like to emphasise that the goal of our paper is to study and propose a novel method for addressing the issue of model misspecification in SBI and is not about addressing the issue of posterior calibration as studied in [1-3]. While we agree the latter issue is important and has not been entirely solved, there exist reasonable solutions to cope with this problem (when ignoring model misspecification), as proposed in [2], [3], or by increasing the simulation budget and optimizing the neural network hyperparameters. Moreover, the miscalibration of a surrogate posterior model can be directly measured and potentially addressed with the aforementioned strategy. Handling model misspecification is a distinct issue as it can only be observed via the gathering of real-world data. Thus our main goal is not to obtain a perfectly calibrated posterior distribution, as we agree this may be arbitrarily difficult even in the well-specified SBI setting. Instead, we aim to create predictive models that 1) exploit the information present in the simulator, even if it is misspecified, 2) are informative (i.e., do not simply output the prior distribution), and 3) are reasonably well calibrated, such that the predicted uncertainty is not overly overconfident.
>
> As a remark, in Appendix F, we do not assume the SBI model is calibrated but “marginally calibrated”, which simply means the expectation of the posterior distribution over the marginal distribution of the observation matches the prior (as defined by eq 9). This property is arguably easier to check and might also be easier to ensure than the standardly checked average “per-observation calibration”.
> From your question, we understand our phrasing might have been misleading and wrongly interpreted as a claim to solve the issue of calibration in SBI. We will ensure to add the necessary context for the reader to understand why we highlight this “self-calibration” property.
>
> [1] Hermans et al. (2022). A Crisis In Simulation-Based Inference? Beware, Your Posterior Approximations Can Be Unfaithful. TMLR.
>
> [2] Falkiewicz et al. (2023). Calibrating Neural Simulation-Based Inference with Differentiable Coverage Probability. NeurIPS.
>
> [3] Delaunoy, Arnaud, et al. "Towards reliable simulation-based inference with balanced neural ratio estimation." Advances in Neural Information Processing Systems 35 (2022): 20025-20037.
>
> **Looking at how the authors ... instead of NPE-RS and NNPE.**
>
> We believe your confusion might stem from some language issue caused by the double use of the same word “calibration” that we make along the paper and refers to the calibration of the posterior distribution and is also used to denote the set of labelled data RoPE uses to address misspecification. Nevertheless, we believe our work is about addressing the issue of model misspecification, instead of posterior calibration. Indeed, the posterior is a direct by-product of the model, which we see as defining the joint distribution between parameters and observations, and when we define misspecification as a gap between two posterior distributions, this can be equivalently seen as a gap between two generative processes: the one described by the simulator with a prior distribution and the one observed in the real world. Applying calibration methods such as introduced in [1] and [2] would not solve the problem we are interested in as these methods only focus on the simulator and do not consider potential gap between real-world and simulated samples. Our experiments also focus on such “misspecification scenarios” as exemplified by Task E, where the data simulated and real-world measurements are drastically different. We agree that under no model misspecification, it could be interesting to study whether RoPE can help improve the calibration of NPE; however, this is not the goal we target in the paper. We believe the second paragraph of the introduction provides an explicit mention of use cases we are interested in and why these are related to model misspecification. We do not believe [2] will provide a viable solution in our context as the method relies on Monte Carlo estimate of model calibration, which itself necessitates large sets of labelled data. [1] and [2] use the simulator to generate such labelled data. In contrast, we can only real-world labelled data, which are more expensive to collect. Extending these methods to handle model misspecification as we define it is an open research direction.

---

> ### Author Response · Authors · 2024-11-15
> **Authors' response [cont.]**
>
> **The equation in line 190 ... and the implications for when it does not hold.**
>
> The main difference is that NNPE is very restrictive on the class of model misspecification, whereas RoPE can theoretically learn to be robust to any model misspecification that respects eq 3. The spike and slab model considered in NNPE is very restrictive and does not provide a viable solution for many real-world use cases where a practitioner would like to exploit domain knowledge in the form of a simulator to infer the properties of a real-world system. We observed this weakness in our empirical results (solid orange line in in Figure 2). It is, for instance, pretty clear that x_s and x_o in Task E are not related to each others via a spike and slab model. The equation in line 190 is correct if and only if eq. 3 is satisfied. Indeed,
> $$ p(\theta \mid x_o) = \int p(\theta, x_s \mid x_o) dx_s = \int p(\theta \mid x_s) p(x_s \mid x_o) dxs  $$
>
> $$ \iff  p(\theta \mid x_s) =  p(\theta \mid x_s, x_o) $$
>
> $$ \iff p(x_o, \theta \mid x_s) = p(x_o \mid x_s)  p(\theta \mid x_s, x_o)  =   p(x_o \mid x_s)  p(\theta \mid x_s) $$
>
> $$ \iff x_o \perp \theta \mid x_s $$
> by definition of conditional independence and following Bayes rule. Thus NNPE, although it does not explicitly state it, makes the same assumption. Thus this modelling assumption is necessary to make the factorization we use correct. If the assumption is wrong, there might be additional information flowing from $\theta$ directly to x_o that our predictive model would not be able to exploit. This assumption is a modelling assumption and its relevance is thus problem dependent. Overall we believe that a well-designed simulator should represent a robust relationship between the parameters and the observation and thus using this assumption should be beneficial to make predictions that are robust to environment / distribution shifts, as we explain in lines 192 to 200.
>
> **The notation is unclear at times ... which is not introduced.**
>
> In lines 81-90 we use the notation $\mathbb{P}^\star$ to introduce the standard definition of model misspecification, as is usually formalized in the literature. From line 90 on we are introducing our extended notation for model misspecification, whose necessity we justify also with the example in Appendix B. $p^\star(\theta \mid x_o)$ is the density of the “true” posterior distribution of the parameter given an observation in the real world. This follows implicitly from $p^\star(\theta)$ and $p^\star(x_o \mid \theta)$ that are introduced to represent the “true” marginal distribution of parameters and “true” conditional distribution of observation given the parameter value.
>
> **In section 2.2 ...  to understand every detail.**
>
> We could add the following intuition:
> An optimal transport (OT) coupling is a mathematical object that represents the most efficient way to associate two probability distributions while minimizing a cost function that measures the "distance" between samples drawn from one distribution and their counterparts in the other.
>
> **Minor comments:**
>
> Indeed, thank you for noticing these typos.

---

> > ### Author Response · Authors · 2024-11-15
> > **Authors' response [cont.]**
> >
> > **In section 3.1 ... Are there any guarantees?**
> >
> > We do not believe that we can guarantee that optimizing equation (6) will necessarily yield sufficient statistics. However, we know that real-world measurements labelled with a certain parameter value should ideally be associated with simulated samples corresponding to the same parameter during the OT coupling. By minimizing the L2 norm between the average sufficient statistics on these synthetic samples and the one predicted for the real-world measurement, we ensure the cost of associating these samples together is small. Our empirical results demonstrate that the chosen strategy performs well on a large set of diverse tasks. Nevertheless, we believe investigating alternative fine-tuning strategies is an exciting research direction.
> >
> > **In most of the experiments ... very simple problems?**
> >
> > We rather believe these results demonstrate that combining a supervised fine-tuning step together with and OT step, which ensure the observed and simulated data marginal distributions are matched to eachother, is an effective way to exploit both supervised and unsupervised data. Nevertheless, RoPE is not perfect and there may be multiple for the performance to quickly saturate. Indeed, if the simulator is extremely good or bad, it may respectively make the issue of misspecification very easy or impossible to address by RoPE. In this case, we would observe a plateau. In addition, the RoPE fine-tuning strategy is simple and may be suboptimal in cases where the calibration set size is larger, explaining why performance plateaus on some benchmarks. Nevertheless, we observe that RoPE systematically outperforms baselines, yielding well calibrated and informative posterior distributions. The RoPE framework could be improved for particular use cases by adapting the fine-tuning step and ensure it extracts as much information as possible from the real data, while aligning them with the sufficient statistics from the simulated data.

---

> > > ### Comment · Reviewer_YfYx · 2024-11-20
> > >
> > > Thank you for your response. Here are some of my follow-up thoughts:
> > >
> > > * **``...when we define misspecification as a gap between two posterior distributions, this can be equivalently seen as a gap between two generative processes: the one described by the simulator with a prior distribution and the one observed in the real world.''**: I think my confusion stems from this definition of model misspecification. Correct me if I am wrong, but I see three different cases that could lead to "model misspecification" as you define it:
> > >     1. When the true data generating process does not belong in the family of distributions induced by the simulator.
> > >     2. When the ``true prior'' does not match the prior used during SBI.
> > >     3. When there is no mismatch between the true DGP and the simulator, and the priors, but the inference procedure yields a posterior that does not match the true posterior (either because it introduces bias or does not provide accurate uncertainty quantification). It was due to this 3rd case that I talked about calibration problems in SBI
> > >
> > >     I believe you are focused on the first case. The premise being that the simulator actually is ``well-specified'' (in the sense that it models the correct relationship between $x$ and $\theta$ that occurs in the real-world), but the observed data $x_o$ comes from some noisy transformation of $x_s$: $p(x_o | x_s)$ (similar to Ward et al. 2023). Now, we can try to assume some form of this noise model and learn it (e.g. a spike-and-slab), but that's not going to be true in every case. So if we could just have some additional $(x, \theta)$ pairs (which you call calibration data) that is not corrupted by any noise, we can use this to learn features (or statistics) that are `ìnvariant'' to this noise. If you agree with this summary of your work (and please let me know if you don't), then my feedback is that this does not come across clearly from the current version of the paper.
> > >
> > >     My question is then what about cases 2 and 3? Seems like you are assuming that the prior used in RoPE *is* the true prior $p^\star(\theta)$ (line 130). This seems like a really strong assumption to me, and I am not sure how does one satisfy this in practice (I am not even going into the philosophical discussion of what does it even mean to have a *true prior in the real-world*). Moreover, there seems to be little to no discussion about this assumption in the paper, the only exception being the ``prior misspecification'' experiment, where the prior is not exactly misspecified in my opinion, just slightly different. In the end, the confusion is that your definition encompasses more cases but you use it to mean only one thing.
> > >
> > > * **Regarding notation clarity**: Thank you for clarifying in your response what the notation means. However, my comment was that they are not introduced properly in the paper. Plus, you did not answer my question regarding the difference between $\mathbb P^\star$ and $p^\star(x_o)$.
> > >
> > > * **...if the simulator is extremely good or bad, it may respectively make the issue of misspecification very easy or impossible to address by RoPE**: I don't understand this comment. Can you please clarify? So in your experiments, which case was it that we see a plateau in the results?

---

> ### Author Response · Authors · 2024-11-20
> **Answers to follow up questions**
>
> **Model misspecification & introducing $p^\star(\theta \mid x\_o)$**
>
> Thanks a lot for your follow up thoughts, they really helped us understand your very valid concerns. We have reformulated our definition of model misspecification (lines 73-78) in an equivalent way that illuminates these points better. We write our new definition here, and address your comments below.
> [New definition start]
> First, we assume the calibration set $\{(\theta^i, \mathbf{x}\_o^i)\}\_{i=1}^{N\_c}$ of real-world observations $\mathbf{x}\_o \in \mathcal{X}$ and their corresponding labels $\theta \in \Theta$ are sampled i.i.d. from a joint distribution given by the density $p^\star(\theta,\mathbf{x}\_o)$. Let $p^\star(\theta)$ be the marginal density of the underlying parameters $\theta$ in the real world, and $p^\star(\mathbf{x}\_o) := \int_\Theta p^\star(\theta) p^\star(\mathbf{x}\_o \mid \theta) d\theta$ be the marginal density of the real-world observations, where $p^\star(\mathbf{x}\_o \mid \theta)$ is the unknown process which is modeled by the simulator $p(\mathbf{x}\_s\mid \theta)$. We say the simulator is misspecified if $\exists \mathcal{S} \subseteq \Theta\times\mathcal{X}$ with $\iint\_\mathcal{S} p^\star(\theta, \mathbf{x}) d\theta d\mathbf{x} > 0$ such that $p^\star(\mathbf{x}\_o\mid\theta) \neq p(\mathbf{x}\_s=\mathbf{x}\_o\mid\theta)$.
> [New definition end]
> This is equivalent to the old definition, if we had explicitly introduced $p^\star(\theta \mid x\_o)$ and $p(\theta \mid x\_o)$ (like you suggested) as:
> $$p^\star(\theta \mid \mathbf{x}\_o) := \frac{p^\star(\theta)p^\star(\mathbf{x}\_o \mid \theta)}{p^\star(\mathbf{x}\_o)}$$
> and
> $$p(\theta \mid \mathbf{x}\_o):= \frac{p^\star(\theta) p(\mathbf{x}\_s = \mathbf{x}\_o \mid \theta)}{p^\star(\mathbf{x}\_o)},$$
> which we agree should have been more explicit in our original definition.
>
> In light of this reformulated definition, let us discuss the reasons for misspecification (1-3) that you outline.
> Regarding reason (1): note that our definition is stronger: even when the true data generating process does belong in the family of distributions induced by the simulator, the simulator may still be misspecified under our definition (as we showed in the example in Appendix B), because we require that for a fixed value of $\theta$---instead of “some” value of $\theta$---the simulator matches the distribution of the real data. Is this clearer now?
>
> Regarding point (2): under this definition, if the simulator is misspecified it will produce the wrong distribution *even when the prior used for inference is the true prior*.
>
> Regarding point (3): indeed, even without misspecification, the estimated posterior can be misscalibrated due to the inference tool (e.g., NPE) that one employs, as shown in the references you suggested. However, this point is orthogonal to the above definition, i.e., it is a problem of the estimator even when the simulator is correctly specified. In our definition, we do not talk about the estimation of the posterior but consider its “oracle” form as a by-product of the simulator and the prior distribution. We understand your confusion and would like to clarify this more explicitly in the revised version of the manuscript. Do you think this is reasonable?

---

> > ### Author Response · Authors · 2024-11-20
> > **Answers to follow up questions [cont.]**
> >
> > **[cont. model misspecification & introducing $p^\star(\theta\mid x_o)$]**
> >
> > Your summary of our method—learning some invariant features between synthetic and real observations—is correct, although we didn’t fully understand what you mean by saying that the calibration data is “not corrupted by any noise”. Our calibration is rather composed of real-world, potentially noisy, measurements $\mathbf{x}\_o$ paired with their ground truth underlying parameter value $\theta$, obtained by manipulating the system considered or via an independent and trusted measurement process. Nonetheless, you have a good point and we can make this “summary” more explicit, e.g., in the discussion.
> >
> > Now, regarding your point about the **true prior** and its effect on the method. We agree with avoiding the philosophical discussion about what a true prior means. Instead, we can look at the prior used for inference from an algorithmic view: it controls which synthetic observations are seen during training. These influence the statistics we learn with the NSE $\mathbf{h}\_{\omega^\star}$, which we then fine-tune into $\mathbf{g}\_{\phi}$ for the real observations with the calibration set. Because we use unbalanced OT during inference, the method is not severely affected by a misspecified prior (as we wanted to show in Figure 4), as long as they share most of the support. Naturally, if the prior is misspecified in a way that certain values of $\theta$ are never seen during training, it is safe to say that the method will likely fail. But isn’t this to be expected of any SBI, or Bayesian infefrence, method? Even from a frequentist point of view, one must decide on the domain of the parameters for any inference method to be sensible. Furthermore, in many real-world applications—e.g., the cardiovascular and monitoring examples in the introduction—even if we can’t set a “correct” prior, it is reasonable that a validity range for the target parameters is known, which is what motivated the setting in Figure 4.
> >
> > Are things more clear now with the new definition? To summarize:
> > - Our definition focuses on a stronger version of reason (1)
> > - The construction of our method offers some robustness to reason (2)
> > - Reason (3) is always present when we use SBI and, while (very!) important, is orthogonal to the whole discussion about misspecification (which we hope is clear with the reformulated definition)
> >
> > **Difference between $\mathbb{P}^\star$ and $p^\star(x_o)$**
> >
> > Sorry about not answering your question. So, both $\mathbb{P}^\star$ and $p^\star(x_o)$ represent the marginal distribution (density) of the real observations. We use two notations because in the usual definition of misspecification that we introduce in lines 82-85 (see also references there), the setup is that this marginal distribution comes from a single value of the parameter $\theta$. In our case we assume there is instead some underlying distribution $p^\star(\theta)$ over the parameters $\theta$ that give rise to $p^\star(x_o)$ through the equation in line 95/96. We believe it is feasible to unify the two definitions under some additional assumptions on what kind of models we consider and which interpretations we give to its parameters however we felt it would be more natural to have a different definition that rather relates explicitly to the goal of inferring value of theta "correctly" -- instead of being centred around the generative model for the observations $\mathbf{x}$. Does this answer your question about $\mathbb{P}^\star$ and $p^\star(x_o)$?

---

> > > ### Author Response · Authors · 2024-11-20
> > > **Answers to follow up questions [cont.]**
> > >
> > > **...if the simulator is extremely good or bad, it may respectively make the issue of misspecification very easy or impossible to address by RoPE: I don't understand this comment. Can you please clarify? So in your experiments, which case was it that we see a plateau in the results?**
> > >
> > > We meant that when the model is almost perfectly specified (i.e., misspecification is minimal), the fine-tuning step should quickly converge to an optimal solution that generalizes well beyond the fine-tuning set. In this case, increasing the size of the calibration set is unlikely to significantly impact the final solution. Conversely, when the model is highly misspecified, the fine-tuning strategy may struggle, and we would not expect significant improvements from increasing the calibration set size. In this extreme scenario, RoPE might end up reverting to the prior distribution—but this behavior was not observed in our experiments.
> > >
> > > In the case of minimal misspecification, RoPE's performance might be suboptimal depending on the choice of the entropy-control parameter, $\gamma$. For higher values of $\gamma$, RoPE explicitly aims to produce under-confident posteriors. We believe the plateau observed in the log-predictive probability (LPP) for some tasks is likely related to this issue. Adjusting $\gamma$ could shift this plateau, but in our experiments, we did not fine-tune $\gamma$ for each benchmark. This was a deliberate choice to honestly report results, as we believe that optimizing $\gamma$ in practice is problem-dependent.
> > >
> > > The key takeaway from our results is that, even without fine-tuning $\gamma$, RoPE produces posteriors that are more informative and better calibrated than state-of-the-art methods for addressing model misspecification in SBI. That said, we acknowledge that in practical applications, adapting the fine-tuning step and optimizing $\gamma$ may be crucial for achieving the best results.
> > >
> > > We view RoPE as a general framework rather than a fully automated solution to model misspecification. Addressing this challenge effectively requires iterative refinements, combining domain expertise with experimentation in real-world applications.

---

> > > ### Comment · Reviewer_YfYx · 2024-11-22
> > >
> > > Thanks for your response. We are getting somewhere now.
> > >
> > > * **...we didn't fully understand what you mean by saying that the calibration data is “not corrupted by any noise”:** Oh yes you are right. The observed data $\mathbf{x}_o$ is always corrupted, it's just that you have the ground truth $\theta$ for a few of those, correct? What I don't understand now is what you mean by "ground-truth parameters". $p^\star(\mathbf{x}, \theta)$ is the true joint density in the real-world, so a point $(\mathbf{x}_o, \theta)$ in the calibration set is a sample from this true joint?
> > >
> > > * **But isn’t this to be expected of any SBI, or Bayesian inference, method?:** For neural SBI methods yes. But when you are doing likelihood-based Bayesian inference, prior misspecification manifests as a data-prior conflict, and as you collect more data, the effect of prior misspecification increases. For neural SBI methods, if your prior places negligible mass on the true $\theta$, then no amount of additional observed data is going to help you, as your network would always view it as out-of-distribution.
> > >
> > > * **Difference between $\mathbb{P}^\star$ and $p^\star(x_o)$:** Usually people denote the distribution by upper case letters (like $\mathbb{P}^\star$) and the corresponding density by lowercase $p$. Here, you are abusing notation a bit by using lowercase for distributions and density both, which is not a huge problem as long as you are consistent. I would suggest just removing $\mathbb{P}^\star$ to avoid this confusion, and just stick to the lowercase $p^\star$.
> > >
> > > (I need to think a bit about your new definition above, but I am running out of time. Will get back to you on it as soon as I can.)

---

> > > > ### Author Response · Authors · 2024-11-22
> > > >
> > > > Thank you for your prompt response. We are happy to hear that! We answer your points below. Please let us know if anything is unclear in our answer.
> > > >
> > > > **ground truth $\theta$ for a few of those, correct?... so a point $(\mathbf{x}_o, \theta)$ in the calibration set is a sample from this true joint?**
> > > >
> > > > Yes! That’s correct. Exactly, a point $(\mathbf{x}_o, \theta)$ is a sample from this true joint (and $\theta$ is the “ground-truth” parameter for the corresponding $\mathbf{x}_o$).
> > > >
> > > >
> > > > **But isn’t this to be expected of any SBI, or Bayesian inference, method?**
> > > >
> > > > We fully agree that this is an issue in neural SBI methods. We hope that the new definition made the distinction between simulator and prior misspecification more clear.
> > > >
> > > > **Difference between $\mathbb{P}^\star$ and $p^\star(x_o)$**
> > > >
> > > > That’s a good point. We will do that, thanks.

---

> > ### Comment · Reviewer_YfYx · 2024-11-27
> > **On the new definition**
> >
> > Apologies for the delay in getting back to this. I have been thinking about your new definition, and frankly, I am only getting more confused. Here are my thoughts:
> >
> > * Firstly, it is good that now misspecification is defined wrt the simulator and not the posterior. However, I do not see that the new definition is equivalent to the previous one, as the previous one includes $p^\star(\theta)$. Moreover, I think your expression for $p(\theta | \mathbf{x}_o)$ is not correct: its denominator shouldn't be $p^\star(\mathbf{x}_o)$, but whatever you get by marginalizing out $\theta$ from the numerator expression.
> >
> > * I am trying to understand the new definition, but I don't see how it is different from the misspecification definition that Ward et al. 2023 and Huang et al. 2023 use, except that your true data generating distribution is also conditioned on $\theta$. Maybe you can give a simple example where we can take $p^\star(\mathbf{x}_o | \theta)$ to be $\mathcal{N}(\theta, 1)$, and so on to get some intuition about this definition?
> >
> > * I am starting to wonder what is even the point of defining misspecification this way. Where in your method do you actually use it? Seems like the important thing is the assumption of conditional independence given simulated data in equation 3. (I could just have forgotten or missed the point so please bear with me.)
> >
> > * So you say $p^\star(\mathbf{x}_o , \theta) = p^\star(\mathbf{x}_o | \theta) p^\star(\theta)$, but you also assume that $p(\mathbf{x}_o, \theta) = p(\mathbf{x}_o | \mathbf{x}_s)p(\theta | \mathbf{x}_s)$. Can you again give a toy example where you assume everything to be simple Gaussians or something where these things hold?
> >
> > * Since you observe both $\theta$ and $\mathbf{x}_o$ in the real world, we can take the "data" to be $\mathbf{y}_o = (\mathbf{x}_o, \theta)$, with the new data space being $\mathcal{Y} = \mathcal{X} \times \Theta$, and $p^\star(\mathbf{y}_o)$ being the true data distribution. Now then the simulator can be $p(\mathbf{y}_s | ?) = p(\mathbf{x}_s , \theta | ?) = p(\mathbf{x}_s | \theta, ?) p(\theta | ?) p(?)$. Here, the question mark (?) would be the latent variables that you do not observe in the real world but are needed for simulating either $\mathbf{x}_s$ or $\theta$ or both. In your case, there is no ?, but $\theta$ is both the latent variable and something you can observe. I would like to hear your thoughts on if any of this makes any sense to you (or am I rambling nonsense).

---

> ### Author Response · Authors · 2024-11-27
>
> Thanks for the further questions and remarks.
>
> **Typo in the numerator:**
>
> You are correct; there was a typo in our previous response regarding the numerator for the posterior's simulator. It should read:
> $$p(\theta \mid x\_o) := \frac{p^\star(\theta) p(x\_s = x\_o \mid \theta)}{\int p^\star(\theta) p(x\_s = x\_o \mid \theta) \text{d}\theta}$$
> Apologies for the confusion. As a note, this typo is not present in the updated manuscript.
>
> **Reasons to use a different definition:**
>
> You are right to point out that $\theta$ plays a key role in our definition. Unlike the misspecification definition that Ward et al. 2023 and Huang et al. 2023 use, we assign a clear interpretation to $\theta$ (often as a physical quantity) and aim to preserve it even under misspecification. We achieve this by assuming the calibration set provides trustworthy labels, as is realistic for many real-world scenarios. Appendix A2 provides an example highlighting this distinction. For instance, if the true generative model is $\mathcal{N}(\theta, 1)$ but the simulator assumes $\mathcal{N}(\theta + \alpha, 1)$ (where $\alpha$ is the misspecification), traditional definitions might consider this model well-specified, whereas our approach accounts for the semantic importance of $\theta$. As you noted in your last comment, we agree that it is possible to reconcile the two definitions by re-stating what are the model parameters and observations.
>
> The goal of re-stating the definition is to clarify the settings we are interested in: predicting $\theta$ accurately (so that it can be used for downstream applications/decisions that assumes $\theta$ has a certain meaning), even when the simulator is misspecified. With our definition, we also discard prior misspecification and instead focus on issues with the simulator (though we also evaluate robustness of our algorithm in presence of prior misspecification). In contrast, eq. (3) in the paper is a useful modeling tool—even if imperfect—that motivates RoPE and whose positive and negative consequences are discussed in the paper.
>
> **Distinguishing assumptions:**
>
> The first assumption, as long as $p^\star(\theta)$ and $p(x\_o \mid \theta)$ are properly defined, is inherently valid. In contrast, the second assumption (no direct dependence between $\theta$ and $x\_o$ beyond what is modeled by the simulator) is a practical modeling simplification. We can provide an example where this assumption holds and another where it does not, to clarify its scope:
> #### Example where eq 3 is respected:
> $$\theta  \in \mathbb{R} \sim p^\star(\theta)  \quad x\_s \in \mathbb{R} \sim \mathcal{N}(\theta, 1) \quad x\_o = x\_s - \alpha \iff x_o   \in \mathbb{R} \sim \mathcal{N}(\theta - \alpha, 1)$$
> #### Example where eq 3 is **not** respected:
> $$\theta  \in \mathbb{R} \sim p^\star(\theta) \quad x\_s \in \mathbb{R}^2 \sim \mathcal{N}([\theta, 0]^T, I) \quad  z \sim \mathcal{0, 1} \quad x_o = x_s + [0, \theta * z]^T  - \alpha  \in \mathbb{R}^2 \iff x_o  \sim \mathcal{N}([\theta, \theta] - \alpha, 1)$$
>
> **Reconciling perspectives:**
>
> This is a great point and we also considered presenting the perspective of treating $(x\_o, \theta)$ as a joint observation to reconcile the two definitions. However, we believe this perspective obscures the distinct roles of $x$ (observed at test time) and $\theta$ (the quantity to predict) that is important in our setting. We were afraid bringing this into the discussion might be more confusing and this is why we avoided it but we also appreciate this dual view and we would be glad to incorporate this perspective in Appendix A2 if you think it would be helpful.
>
> Please let us know if you have further questions! Thanks a lot for the very constructive discussion.

---

> > ### Comment · Reviewer_YfYx · 2024-11-27
> > **Final thoughts**
> >
> > Thank you for the quick response. I will summarize my thoughts now and won't take up more of your time.
> >
> > I think anything that helps the reader understand your setting better would be nice to include, including these simple cases, dual view, etc. that we have been discussing. Otherwise you get readers like me who just get thrown off by the definition an can't move past it.
> >
> > Unfortunately, I won't be increasing my score, as I don't think the paper is ready yet (our entire back and forth about the definition is a signal for this). Apart from that:
> >
> > * The paper does not properly motivate why using optimal transport is the right thing to do. Given your setting/definition/assumptions, there could potentially be many possible solutions to this problem. Why is the proposed solution the right one? I understand that you *can* model misspecification with OT, and doing so helps in the examples you have in Section 4. But the paper directly jumps from the problem setup to the solution without telling where the solution came from.
> >
> > * There needs to be more discussion regarding the hyperparameter and how to set it in practice. At best, you should provide an algorithm, or at the least some recommended guidelines/heuristics based on analyzing its effect.
> >
> > Addressing the above two points would then help improve the technical contribution of the paper, which currently can be thought of as incremental (the application of OT in this setting).
> >
> > I apologize for being the kind of reviewer that asks a million questions but ends up saying "I keep my score". Clearly, some of the other reviewers do not feel the same as me, and so I wish you all the best!

---

> ### Author Response · Authors · 2024-11-27
> **Final considerations**
>
> > **Thank you for the quick response. I will summarize my thoughts now and won't take up more of your time.**
>
> We are very grateful for this interaction, for having spent so much time reading our rebuttal, and overall, for your interest in this discussion.
>
> > **I think anything that helps the reader understand your setting better would be nice to include, including these simple cases, dual view, etc. that we have been discussing. Otherwise you get readers like me who just get thrown off by the definition an can't move past it.**
>
> We seized the opportunity given by this rebuttal to improve, through what we believe are a few minor cosmetic modifications, the clarity of our paper. This is in part thanks to you, and the time you have spent reading, understanding and criticizing our paper.
>
> > **Unfortunately, I won't be increasing my score, as I don't think the paper is ready yet (our entire back and forth about the definition is a signal for this).**
>
> We of course respect your opinion. We have benefited from this interaction nonetheless, and we believe the paper is better now than it was when first submitted.
>
> > **The paper does not properly motivate why using optimal transport is the right thing to do. Given your setting/definition/assumptions, there could potentially be many possible solutions to this problem. Why is the proposed solution the right one? I understand that you can model misspecification with OT, and doing so helps in the examples you have in Section 4. But the paper directly jumps from the problem setup to the solution without telling where the solution came from.**
>
> We acknowledge this criticism, and agree that a proper justification on why OT is an effective solution only arrives fairly late in the paper (Section 3.2). We understand this concern and have clarified the motivation for using OT earlier in the text (**L.94–98** and **L.187–188**) to guide the reader more effectively, before they reach section 3.2.
>
> > **There needs to be more discussion regarding the hyperparameter and how to set it in practice. At best, you should provide an algorithm, or at the least some recommended guidelines/heuristics based on analyzing its effect**.
>
> We argue that RoPE has a fairly limited set of hyperparameters (entropic regularization $\gamma$ and unbalancedness regularization $\tau$), which are very classic in all regularized OT works (see for instance [1-5]). Our parameter choice for both has been guided by the extensive literature and software implementations that exist (here, in particular, OTT-JAX [6]). We use the _same_ fixed values of $\gamma$ and $\tau$ throughout the 6 benchmarks (**L.406-410**). With these settings, RoPE performs consistently well across all benchmarks and outperforms NNPE and RNPE. We believe that this is more convincing, from a practitioner's perspective, than a hyperparameter search that would need to be tuned for each experiment.
>
> Furthermore, we encourage you to check in more detail the several experiments in which we study the effect of these parameters (lines 463–508, as well as Figure 3 (right panel) and Figure 4). To assist practitioners, we explicitly provide advice on how to set these parameters within the manuscript: **L.470–473** for $\gamma$ and **L.485–508** for $\tau$. We hope this addresses your concern and offers useful guidance for practical applications.
>
> Again, regardless of your final score, we would like to thank you for the time you have spent reading, criticizing and helping us improve the submission.
>
> [1] Peyré, Gabriel, and Marco Cuturi. "Computational optimal transport: With applications to data science." Foundations and Trends® in Machine Learning 11.5-6 (2019): 355-607.
>
> [2] Genevay, Aude, Gabriel Peyré, and Marco Cuturi. "Learning generative models with sinkhorn divergences." International Conference on Artificial Intelligence and Statistics. PMLR, 2018.
>
> [3] Fatras, Kilian, et al. "Unbalanced minibatch optimal transport; applications to domain adaptation." International Conference on Machine Learning. PMLR, 2021.
>
> [4] Pham, Khiem, et al. "On unbalanced optimal transport: An analysis of sinkhorn algorithm." International Conference on Machine Learning. PMLR, 2020.
>
> [5] Chizat, Lenaic, et al. "Unbalanced optimal transport: Dynamic and Kantorovich formulations." Journal of Functional Analysis274.11 (2018): 3090-3123.
>
> [6]: Cuturi, Marco, et al. "Optimal transport tools (ott): A jax toolbox for all things wasserstein." arXiv preprint arXiv:2201.12324(2022).

---

### Author Response · Authors · 2024-11-26
**Summary of the discussion and updates made to the paper**

We would like to thank the Reviewers and the Area Chairs for their time and effort assessing our submission. We are grateful for the many encouragements we have received for all three main evaluation criteria:
- Soundness:
     - “The paper addresses a relevant practical problem in the field of SBI.”(YfYx)
     - “The paper contributes an interesting approach to a hard problem in SBI and scientific modeling in general.” (6VTN)
     - "The OT formulation and ensemble-based correction are well motivated and the assumptions are clearly stated." (6VTN)
     - "clearly state the contributions and the specific SBI setting the method is designed to solve." (hywB)
     - "The performance of the proposed method is promising and the ablation and hyperparameter sensitivity studies help understanding the method and tend to make it easier for practitioners to choose parameter in practice." (hywB)
     - "Overall, I believe this could be a valuable contribution for the field and I am tentatively recommending to accept this paper. Model misspecification is an urgent problem in SBI and methods for addressing it have been scarce so far. The proposed method and additionally the proposed benchmarking tasks are a promising next step." (hywB)
     - "The paper investigates an important issue." (MKFY)
- Contribution:
     - “The idea of using the optimal transport coupling for learning statistics is neat” (YfYx)
     - "the experimental evaluation seems thorough." (YfYx)
     - "The evaluation in the setting of low-dimensional models is comprehensive and features good comparisons. The results are encouraging even with a very small calibration set." (6VTN)
     - "The paper uses two existing benchmarks and introduces four new model misspecification benchmarks, which is a valuable contribution for the community." (hywB)
     - "Detailed examples and ablation studies are performed." (MKFY)
- Presentation:
     - "The paper is largely well-written" (YfYx)
     - "The paper is well-written and well-structured." (hywB)
     - "The goals of this paper are clearly explained" (MKFY)


We would like to thank reviewers again for their constructive criticism as it has certainly helped us to improve the submission. In particular, we would like to thank Reviewers **MKFY** and **hywB** for showing appreciation for our rebuttal and increasing their scores from $5\rightarrow 6$ and $6\rightarrow 8$ respectively.

## Changes to the manuscript

We have taken into consideration each of the reviewers’ comments to produce an updated version of the manuscript (changes highlighted in blue). Here is a summary of the changes/additions.
1. We have emphasized the distinction between our work and the issue of posterior mis-calibration in well-specified SBI settings as discussed with reviewer YfYx (in lines 41-43 and 758-783).
2. We have improved the definition of misspecification to make it more clear (lines 83-90).
3. We have  extended the discussion of related work with the suggestions from reviewer 6VTN (in the limit of the 10-page constraint) in lines 64 to 76.
4. We have **repositioned the figures and made them more readable** following the suggestions of reviewer hywB (i.e., colors, markers, line widths)
5. We have further explained what is an OT coupling (in lines 149-151) as suggested by reviewers YfYx and MKFY.
6. We have included **clarification around equation 2** as discussed with reviewer MKFY (in lines 160 and 166-168).
7. We have provided additional details about the **SBI baseline** and how it is computed (lines 383-384 and Appendix I) as discussed with reviewer hywB.
8. We have further discussed some of our results and provided more context on the baselines and experimental setup to the reader as suggested by reviewer hywB and in light of the discussion with reviewer YfYx (lines 390-392, 405-411, 416-418, 423).


We remain available until the end of the rebuttal period to answer any new question or concern you may have. Thanks again for the time you have spent reading our paper and our rebuttal.

---

### Meta-Review · Area_Chair_TxzL · 2024-12-17

**Metareview:**

This paper proposes RoPE, a robust posterior estimation framework for simulation-based inference (SBI) that addresses the issue of model misspecification. It does so by using a small real-world calibration set of ground-truth parameter measurements and formalizing the “misspecification gap” using optimal transport (OT).

I thank the authors and all the reviewers for their engaging discussions. It seems the reviewers agree the problem of misspecification in SBI is important and the idea of using OT in this setting is novel. Throughout discussions between authors and reviewers, some of the reviewers have increased their scores, while others remain hesitant. The main concerns about this paper relate to (1) the soundness of the approach and the definition of the problem; (2) the motivation for using OT; (3) hyper-parameter setting; (4) scalability and scope of the method and (5) a better experimental evaluation.

There was a lot of back and forth between the authors and Reviewer YfYx about (1) to the point that the reviewer remains unconvinced and I think the paper will need much more clarity on this, and how this ties up to the methodological development. I think (2)-(5) all remain valid concerns. Furthermore, the paper does have a limited scope, as pointed out by the reviewers, there are many problems where labeled real-world parameter measurements are simply not available. Considering all this, I tend to agree with Reviewer YfYx and Reviewer 6VTN that the paper is not ready yet and will benefit from incorporating all the feedback given and addressing the most pressing concerns. Therefore, I recommend rejection.

**Additional Comments On Reviewer Discussion:**

There was a lot of back and forth between the authors and Reviewer YfYx about (1) to the point that the reviewer remains unconvinced and I think the paper will need much more clarity on this, and how this ties up to the methodological development. I think (2)-(5) all remain valid concerns. Furthermore, the paper does have a limited scope, as pointed out by the reviewers, there are many problems where labeled real-world parameter measurements are simply not available. Considering all this, I tend to agree with Reviewer YfYx and Reviewer 6VTN that the paper is not ready yet and will benefit from incorporating all the feedback given and addressing the most pressing concerns. Therefore, I recommend rejection.

---

### Decision · Program_Chairs · 2025-01-22

Reject